# Training as Computation: A Resource-Bounded Theory of Continual Self-Play Learning

## Abstract

We study *training as computation* in a continual self-play setting, where a single reasoning model proposes tasks, solves them, and updates itself using verifiable signals from an external executor–verifier interface. Rather than focusing on one-shot models, we analyze the *process-level* dynamics of learning under explicit resource budgets: each generation step is capped by an output budget, and the executor/verifier operate within bounded working space. Within this framework we (i) formalize a general generator–executor–verifier–buffer loop for continual learning with self-proposed curricula; (ii) provide a *process-level characterization* of expressiveness—the set of functions computable by the evolving loop up to time $t$ matches a corresponding SPACE$[\cdot]$ class determined by the budgets; and (iii) show monotone capability growth under explicit, length-aware exploration schedules and curriculum learnability mechanisms, without assuming non-vanishing exploration or relying on supervised traces. Conceptually, the results separate *capability universality* (as properties of the training *process*) from *alignment and safety* (properties of objectives and verifiers). This positions continual self-play as a principled theoretical framework for understanding data-free improvement under explicit resource budgets.

## 1 Introduction

Self-play has become a dominant paradigm whenever external supervision is scarce or non-existent. AlphaZero surpassed human champions by repeatedly playing itself David Silver & Hassabis (2018); constitutional RLHF frameworks refine large language models by critiquing and rewriting their own responses Bai et al. (2022). Recent surveys formalize self-play as agents interacting with copies or past versions of themselves, situate it within multi-agent RL and basic game theory, and offer unified taxonomies that subsume major algorithmic families Zhang et al. (2025b); DiGiovanni & Zell (2021). Beyond games, self-play has driven advances in protein engineering, language-model safety, and multi-agent decision-making for autonomous vehicles Yi Wang (2023); Cornelisse & Vinitsky (2024); Liu et al. (2025); Fang et al. (2025). Hierarchical and memory-augmented variants further scale self-play to complex, embodied tasks, enabling emergent coordinated strategies without human demonstrations (Zhang et al., 2025a; Sodhani & Pahuja, 2018).

**This paper studies the *computational power of the learning process*.** Prior Turing-completeness results mostly treat *architectures*—e.g., RNNs and modern Transformers with unbounded context (Siegelmann & Sontag, 1995; Jorge Pérez, 2021; Bhattamishra et al., 2020; Graves et al., 2014)—where the simulated program is hard-coded in the weights or input. In contrast, the practical success of self-play suggests a distinct question: *what can a training loop compute?* Recent theory for self-play focuses on sample complexity, convergence, exploration/exploitation trade-offs, and (approximate) Nash guarantees in Markov games (Bai & Jin, 2020; Bai et al., 2020; Lanctot et al., 2017), while emerging safety work frames self-play as an online attacker–defender game (e.g., Self-RedTeam) (Liu et al., 2025) and LLM self-play under limited data (Fang et al., 2025). None of these, however, address the *expressiveness* of the *training loop itself*.

**Our results at a glance.** (i) We provide a **process-level characterization**: under explicit coverage, persistence, and learning assumptions, a broad class of *self-play reasoning agents* realizes

exactly the class of functions computable within the working-memory budget, matching classical SPACE[·] hierarchies. (ii) With an adaptive exploration schedule, we obtain a **finite-time completeness** bound(Appendix D): every function of Kolmogorov complexity $\leq \ell$ is solved with probability $\geq 1 - \delta$ after

$$T(\ell, \varepsilon_0, \gamma, \delta) = O\left(\frac{\ell^{\beta+1}}{\gamma} \ln \frac{1}{\varepsilon_0} + \frac{1}{\gamma} \ln \frac{1}{\delta} + \exp(\ell/c)\right).$$

(iii) Capping the per-step working memory (context window and executor workspace) to $W(t)$ tokens induces a strict **SPACE** hierarchy of solvable tasks:

$$\{\text{functions computed up to time } t\} = \text{SPACE}\big[O(s(W(t)))\big],$$

so if $W_1(t) = o(W_2(t))$ and $s$ is monotone then $\text{SPACE}\big(s(W_1(t))\big) \subsetneq \text{SPACE}\big(s(W_2(t))\big)$ (Thm. 31).

### 1.1 MOTIVATION AND TECHNICAL CHALLENGES

Understanding process-level expressiveness raises challenges atypical of architecture-level proofs. (i) **Stochastic, non-stationary dynamics.** Parameter updates couple generator and solver returns, yielding a non-stationary Markov process that resists classical computability reductions. (ii) **Multi-component alignment.** Correctness requires a pipeline Generator → Executor → Buffer → Solver, with failure modes at any stage. (iii) **Exploration versus specialization.** Universality pressures every finite string $u$ to have non-vanishing probability, yet policy-gradient optimization naturally drives the generator toward high-reward frontier tasks, risking coverage collapse. We resolve this tension through an architectural separation: a mixture of a learned policy-gradient component (for efficiency) and a fixed exploration component (for completeness), ensuring coverage guarantees independently of reward-driven specialization. (iv) **Resource constraints.** Practical systems operate under bounded context/memory and finite wall-clock budgets; we map these budgets to capability classes via a SPACE hierarchy.

### 1.2 CONTRIBUTIONS

We make four contributions.

1. **Process-level universality characterization.** We provide a process-level characterization of a modern self-play training loop: under explicit coverage, persistence, and learning assumptions, the set of functions it can realize coincides with the class $\text{SPACE}[O(s(W(t)))]$ induced by its working-memory budget. This does not improve worst-case computational complexity, but formally links a modern training pipeline to classical space hierarchies.

2. **CPL template.** We introduce a modular *Coverage–Persistence–Learning (CPL)* decomposition: (i) *coverage* supplies a lower bound on the probability of generating any finite program prefix, (ii) *persistence* shows that verified tasks remain permanently accessible under unbounded archival buffers with fair scheduling, and (iii) *learning* proves per-task convergence under unbiased policy gradients in the multi-task interleaved setting.

3. **Finite-time and resource-bounded guarantees.** An adaptive exploration schedule yields the finite-time bound above; capping the per-step working memory (context window and executor space) induces a strict **SPACE**-hierarchy, linking capability growth to resource budgets.

4. **Positioning.** Our buffer-based analysis can be viewed as a PAC-style guarantee *conditional on coverage*, but the overall target class is that of *partial-recursive functions*, well beyond classical PAC regimes and prior self-play theory on sample complexity and equilibria (Zhang et al., 2025b; DiGiovanni & Zell, 2021; Bai & Jin, 2020; Bai et al., 2020).

**Expressiveness versus efficiency.** Our focus in this work is on *expressiveness*: characterizing *which* verifier-defined functions can be realized by a self-play training loop under explicit resource bounds, rather than *how fast* such functions can be learned. Consequently, our finite-time guarantees are exponential in the horizon length, matching the worst-case behavior of enumeration-like search. This should be viewed as a capability upper bound akin to classical results in computability and complexity theory (e.g., universal search and space hierarchies), not as a claim of sample-efficient

learning in the PAC or statistical sense. Improving these worst-case bounds—by exploiting structure in particular problem families or by analyzing when curriculum learning yields polynomial speedups—is an important open direction.

Overall, we shift the focus from architectural universality to *learning-process universality*, providing a reusable template that links exploration schedules, retention policies, and context scaling to formal capability guarantees.

## 2 RELATED WORK

### 2.1 SELF-PLAY IN REINFORCEMENT LEARNING

Comprehensive surveys systematize self-play within multi-agent RL, clarifying preliminaries (Markov games, equilibria) and unifying algorithmic families under a single taxonomy (Zhang et al., 2025b; DiGiovanni & Zell, 2021). Empirically, self-play delivers strong performance across domains: board games, robotics, autonomous driving, and protein design (David Silver & Hassabis, 2018; Cornelisse & Vinitsky, 2024; Yi Wang, 2023). Safety-oriented variants treat self-play as an online attacker–defender game for LLMs (Self-RedTeam; SeRL) (Liu et al., 2025; Fang et al., 2025). On the theoretical side, works analyze sample complexity and equilibrium convergence in competitive RL (Bai & Jin, 2020; Bai et al., 2020; Lanctot et al., 2017), fictitious/self-play variants for imperfect information games (Hu et al., 2023), and automatic curricula (Sodhani & Pahuja, 2018; Sukhbaatar et al., 2018; Raparthy et al., 2020). These results optimize *policy quality* in fixed environments rather than the *computational expressiveness* of the self-play loop.

### 2.2 NEURAL-NETWORK TURING COMPLETENESS

Classical results show that recurrent networks with unbounded precision are universal (Siegelmann & Sontag, 1995). For modern architectures, a line of work proves that Transformers are Turing-complete with unbounded sequence length or memory (Jorge Pérez, 2021; Bhattamishra et al., 2020); differentiable memory systems such as Neural Turing Machines expose an external tape (Graves et al., 2014). These proofs *hard-code* the program in the weights or input and do not address whether a *learning algorithm* can discover such programs via self-play.

### 2.3 BUFFERS, MEMORY, AND RESOURCE–CAPABILITY MAPPINGS

Replay buffers and prioritization schemes improve sample efficiency and stability in RL, while memory-augmented self-play encourages diverse task discovery and faster exploration (Schaul et al., 2016; Sodhani & Pahuja, 2018). Our *persistence* analysis formalizes when unbounded archival buffers with fair scheduling retain the traces required by universal simulation. We further map per-step working memory (context window and executor workspace) to a **SPACE** hierarchy of solvable tasks, providing guidance on buffer capacity in real implementations.

**Distinction.** To our knowledge, there is no prior work connecting *self-play RL* with *Turing universality at the process level*, nor mapping context/buffer limits to classical **SPACE** complexity. Our results complement existing self-play theory (sample complexity, equilibria) and architecture-level completeness by providing the first expressiveness guarantees for the *training loop itself*.

## 3 NOTATION & PRELIMINARIES

We establish the mathematical foundations and key definitions used throughout the paper.

### 3.1 BASIC NOTATION

We work over a finite *token alphabet* $\Sigma$ and write $\Sigma^*$ for the set of all finite strings over $\Sigma$. Boldface denotes random variables; sans-serif denotes computable functions. For any string $u \in \Sigma^*$, we denote its length by $|u|$.

### 3.2 PROGRAMS AND TURING MACHINES

**Definition 1** (Deterministic Python Subset). *Let $\mathcal{L} \subseteq \Sigma^*$ be a syntactically restricted fragment of Python 3 that forbids file I/O, networking and invocation of $\circ s . *$ functions. Running any $p \in \mathcal{L}$ inside the sandbox interpreter $E$ produces a (possibly unbounded) textual output which we denote $E(p, i) \in \Sigma^*$ when $p$ is called with input string $i \in \Sigma^*$. We assume $\mathcal{L}$ is Turing-complete.*

**Definition 2** (Universal Compiler). *Let $TM$ be the set of binary encodings of single-tape deterministic Turing machines. There exists a total computable map*

$$C : TM \longrightarrow \mathcal{L}, \qquad \langle M \rangle \mapsto p_M$$

*such that for every input $w \in \Sigma^*$, the program $p_M$ terminates and prints $M(w)$ iff the machine $M$ halts on $w$.*

### 3.3 COMPLEXITY CLASSES

For a function $s : \mathbb{N} \to \mathbb{N}$, we denote by $\text{SPACE}[s(n)]$ the class of decision problems solvable by a Turing machine using at most $s(n)$ space on inputs of length $n$.

## 4 FORMAL MODEL

We now formally define the self-play reasoning agent architecture and its components.

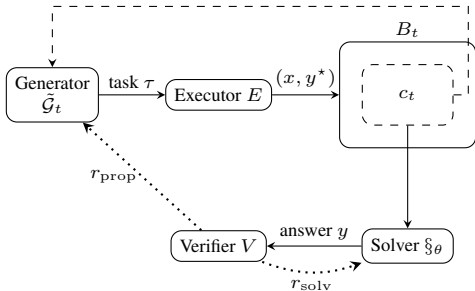

Figure 1: Self-play loop with mixture generator. The effective generator $\tilde{\mathcal{G}}_t$ proposes tasks via the executor $E$ into the archival buffer $B_t$. A bounded working view $c_t \subset B_t$ is sampled for the solver $\S_\theta$ and generator. The verifier $V$ provides distinct rewards $r_{\text{prop}}$ and $r_{\text{solv}}$ to update the generator and solver respectively.

### 4.1 AGENT ARCHITECTURE

**Definition 3** (Generator and Solver Policies). *A self-play reasoning agent is parameterized by weights $\theta \in \mathbb{R}^d$ and consists of two stochastic kernels:*

$$\mathcal{G}_\theta(\,\cdot \mid z, B) : \Sigma^* \times \mathcal{B} \rightsquigarrow \Sigma^*, \tag{1}$$

$$\S_\theta(\,\cdot \mid x, B) : \Sigma^* \times \mathcal{B} \rightsquigarrow \Sigma^*, \tag{2}$$

*called the generator and solver, respectively. Here $z \in \Sigma^*$ is an optional seed, $x \in \Sigma^*$ is a task prompt, and $\mathcal{B}$ is the space of finite multisets of string pairs. Both kernels are neural language models implemented by the same backbone; they differ only in the conditioning prompt.*

*During training, the actual task-generation mechanism is the effective generator $\tilde{\mathcal{G}}_t$, which combines a learned component (optimized via policy gradients) with a fixed exploration component to ensure both efficiency and completeness (Definition 8).*

**Notational convention for parameters.** *In the formal analysis, we conceptually separate the generator and solver parameters, writing $\phi$ for the learned generator component $\mathcal{G}_\phi^{\text{RL}}$ and $\theta$ for the solver policy. In practice, they may share a common backbone with task-dependent heads or be*

*entirely shared with different conditioning; this only strengthens the coupling between task genera-
tion and solving, and does not affect our theoretical arguments. When the distinction is clear from
context, we may write $\theta_t$ to denote the joint parameter state at time $t$.*

**Definition 4** (Verifier). *A* verifier *is defined as the conjunction of polynomial-time decidability and
execution-based verification:*

$$V(x, y) = \min\{V_{\text{poly}}(x, y), V_{\text{exec}}(x, y)\}$$

*where:*

$$V_{\text{poly}}(x, y) = \begin{cases} 1 & \text{if } y \text{ can be poly-time verified for } x \\ 0 & \text{otherwise} \end{cases} \tag{3}$$

$$V_{\text{exec}}(x, y) = \begin{cases} 1 & \text{if } E(p_M, w) = y \text{ and halts within bounds} \\ 0 & \text{otherwise (including non-termination)} \end{cases} \tag{4}$$

*For deduction tasks $\tau_{M,w}$ = <prog>$p_M$</prog><inp>$w$</inp><out></out>, the verifier
returns $V(\tau_{M,w}, y) = 1$ if and only if both:*

1. *The answer $y$ can be verified as correct within polynomial time constraints,* ***and***

2. *Direct execution $E(p_M, w)$ produces output $y$ within the allocated resource bounds.*

*This design ensures that $V(x, y) = 1$ implies $y$ is demonstrably correct through both efficient verifi-
cation and direct computational validation, while maintaining decidability through the polynomial-
time upper bound.*

## 4.2 TASK SPACE AND CLASSES

We adopt three generic task classes: deduction, abduction and induction. Formally let

$$\mathcal{T}_{\text{ded}} = \{(p, i, \bot)\}, \quad \mathcal{T}_{\text{abd}} = \{(p, \bot, o)\}, \quad \mathcal{T}_{\text{ind}} = \{(\{(i_k, o_k)\}_k, \bot)\},$$

where $\bot$ denotes the missing element the solver must infer. All three are finite subsets of $\Sigma^*$ and
therefore share a common super-set $\mathcal{T} \subseteq \Sigma^*$.

In particular, for any TM $M$ and string $w$ we encode the corresponding *induction task*

$$\tau_{M,w} := \text{<inp>}w\text{</inp><out></out>} \in \mathcal{T}_{\text{ind}},$$

whose unique correct answer equals $M(w)$ (verified internally by the generator using $p_M$).

## 4.3 BUFFER AND TRAINING DYNAMICS

**Definition 5** (Task Buffer). *At discrete training step $t \in \mathbb{N}$ the* buffer *is a finite multiset $B_t \subseteq \mathcal{T} \times \Sigma^*$.
Initially $B_0 = \varnothing$. Whenever the verifier accepts a candidate pair $(x, y)$, it is inserted into $B_{t+1}$
either permanently or under reservoir sampling.*

**Definition 6** (Working view of the buffer). *At training step $t$, the agent does not condition on the
full archival buffer $B_t$, but on a bounded-length view*

$$c_t := \psi_t(B_t) \in \Sigma^{\leq W(t)},$$

*where $\psi_t$ is any (possibly stochastic) selection/summarization map satisfying $|c_t| \leq W(t)$.*

*In the resource-bounded setting, all calls to $\mathcal{G}_{\theta_t}$ and $\mathcal{S}_{\theta_t}$ condition on $c_t$ rather than the full buffer:*

$$\mathcal{G}_{\theta_t}(\cdot \mid z, c_t), \quad \mathcal{S}_{\theta_t}(\cdot \mid x, c_t),$$

*although we keep the shorthand notation $B_t$ when the distinction is clear from context.*

**Remark 7.** *Our theory only requires that $\psi_t$ be computable and respect the $W(t)$ bound; practical
implementations are discussed in Appendix E.*

**Definition 8** (Effective Generator with Exploration Mixture). *A self-play reasoning agent consists of a learned generator policy $\mathcal{G}_\phi^{\mathrm{RL}}$ parameterized by $\phi \in \mathbb{R}^{d_g}$ and a solver policy $\mathcal{S}_\theta$ parameterized by $\theta \in \mathbb{R}^{d_s}$.*

*The effective generator $\tilde{\mathcal{G}}_t$ at training step $t$ is defined as a mixture:*

$$\tilde{\mathcal{G}}_t(\cdot \mid z, B_t) := (1 - \epsilon_t)\,\mathcal{G}_{\phi_t}^{\mathrm{RL}}(\cdot \mid z, B_t) + \epsilon_t\,\mathcal{G}^{\mathrm{E}}(\cdot), \tag{5}$$

*where:*

- *$\mathcal{G}_{\phi_t}^{\mathrm{RL}}$ is the* learned generator *optimized via policy gradients to maximize learnability-based proposal rewards (Definition 15).*

- *$\mathcal{G}^{\mathrm{E}}$ is a* fixed exploration distribution *(the "enumerator") with full support over $\Sigma^*$, independent of training state.*

- *$\epsilon_t \in (0, 1]$ is a* mixing coefficient *that governs the exploration rate at time $t$.*

**Remark 9** (Physical interpretation and practical implementation). *While we formalize the generator as an explicit mixture for analytical tractability, in practice this architecture can be realized through a single model with modulated sampling entropy: the exploration component $\mathcal{G}^{\mathrm{E}}$ corresponds to high-entropy sampling (e.g., elevated temperature), while the learned component $\mathcal{G}_{\phi_t}^{\mathrm{RL}}$ corresponds to low-entropy exploitation. This separation of concerns—coverage via $\mathcal{G}^{\mathrm{E}}$ and efficiency via $\mathcal{G}_{\phi_t}^{\mathrm{RL}}$—avoids the tension between universal exploration requirements and gradient-driven specialization.*

**Assumption 10** (Fair access to verified tasks). *With probability 1 (over the randomness of the generator and the scheduler), every verified task $\tau$ that enters the archival buffer is selected for training updates infinitely often.*

*Formally, let $U_t$ be the (possibly stochastic) choice of training task at step $t$. Then with probability 1, for all verified $\tau$ there exists an infinite subsequence $\{t_k\}_{k=1}^\infty$ such that $U_{t_k} = \tau$.*

**Reward specification.** As shown in lines 10–11 of Algorithm 1, the generator and solver are updated using distinct reward signals. Here $r_{\mathrm{prop}}$ denotes the learnability-based proposal reward from Definition 15, while $r_{\mathrm{solv}}$ denotes the external solve reward induced by the verifier (e.g., 1 for a correct solution, 0 otherwise).

Critically, although the *effective generator* $\tilde{\mathcal{G}}_t$ samples from a mixture (Definition 8), the policy-gradient updates in line 10 apply only to the *learned component* $\mathcal{G}_{\phi_t}^{\mathrm{RL}}$, optimizing it to concentrate on capability-frontier tasks (those with intermediate success probability) as characterized by Lemma 16. The fixed exploration component $\mathcal{G}^{\mathrm{E}}$ remains stationary throughout training, providing universal coverage independently of the reward-driven learning dynamics. This architectural separation ensures that gradient-based optimization can specialize the learned generator without compromising the exploration guarantees required for computational universality.

See Remark 24 for the precise update equations.

## 5 METHOD

To establish a process-level characterization of computational universality without drowning in non-stationary dynamics, we decompose the training loop into three orthogonal properties. If these three hold simultaneously, the system realizes the desired function class.

### 5.1 PROOF FRAMEWORK

Our proof strategy decomposes the closed self-play loop into three orthogonal properties—*coverage*, *persistence*, and *learning*—each captured by a dedicated lemma.(Detail proofs C)

**Coverage:** The generator mixture architecture (Definition 8) ensures every finite program-input pair is eventually proposed with probability 1 through the fixed exploration component, independently of learned specialization (Assumption 20, Lemma 26).

**Persistence:** Once a task is verified and enters the buffer under an unbounded-growth policy, it remains permanently accessible to future gradient updates. Combined with the Fair Scheduling Assumption 10, each verified task is selected for training updates infinitely often with probability 1. (For practical finite-capacity variants, see Appendix E; these do not enter the main universality proof.)

**Learning:** Once a task becomes permanent in the archival buffer, unbiased policy-gradient updates on the solver drive its success probability to 1 almost surely (Lemma 28). In parallel, the learnability-based proposal reward (Def. 15) and the Frontier Lemma (Lemma 16) show that the *learned generator component* $\mathcal{G}_{\phi_t}^{\mathrm{RL}}$ is shaped to favor capability-frontier tasks over always-easy or always-hard ones, accelerating the discovery of informative tasks. Together, these mechanisms ensure that learning operates efficiently on both the solver's ability to solve verified tasks and the learned generator's ability to identify frontier tasks, while the fixed exploration component maintains completeness guarantees.

**Remark 11** (Scope of persistence guarantee). *Theorems 25 and 31 use the unbounded-archive regime; finite-capacity variants (Appendix E) provide practical guidance.*

**Remark 12** (Persistence vs. catastrophic forgetting). *The persistence guarantee concerns* data availability *(task retention in buffer), distinct from* catastrophic forgetting *(weight-level interference). Our analysis assumes sufficient capacity for per-task convergence (Learning Lemma 28); detailed discussion of capacity constraints and practical forgetting mitigation appears in Appendix E.*

**Definition 13** (Per-task success probability). *Fix solver parameters $\theta$ and an archival buffer state $B$ with working view $c = \psi(B)$ as in Definition 6. For any task $\tau = (x, y^\star)$ with $\tau \in B$, we define the per-task success probability*

$$p_\theta(\tau; B) := \Pr_\theta[\mathcal{S}_\theta(y^\star \mid x, c) = 1],$$

*where the probability is over the solver's internal sampling (and any stochasticity in $\psi$). When $B$ is clear from context we abbreviate $p_\theta(\tau; B)$ as $p_\theta(\tau)$. At training step $t$ we write*

$$p_t(\tau) := p_{\theta_t}(\tau; B_t).$$

**Definition 14** (Learnability zones). *Fix thresholds $0 < \alpha < \beta < 1$. For given solver parameters $\theta$, the task space is partitioned into three zones according to Definition 13:*

$$
\begin{aligned}
\mathcal{E}_\theta &= \{\tau : p_\theta(\tau) \geq \beta\} && \text{(always-easy zone)}, \\
\mathcal{H}_\theta &= \{\tau : p_\theta(\tau) \leq \alpha\} && \text{(always-hard zone)}, \\
\mathcal{F}_\theta &= \{\tau : \alpha < p_\theta(\tau) < \beta\} && \text{(frontier zone)}.
\end{aligned}
$$

*Tasks in $\mathcal{F}_\theta$ are those that currently provide informative learning signal for the solver.*

**Definition 15** (Learnability-based proposal reward). *Given solver parameters $\theta$ and thresholds $0 < \alpha < \beta < 1$, the learnability-aware proposal reward for a task $\tau$ is defined as*

$$r_{\mathrm{prop}}(\tau; \theta) := \mathbf{1}\{\tau \in \mathcal{F}_\theta\}.$$

*More generally, one may use any smooth surrogate that vanishes on $\mathcal{E}_\theta \cup \mathcal{H}_\theta$ and is maximized on $\mathcal{F}_\theta$, such as*

$$r_{\mathrm{prop}}(\tau; \theta) = p_\theta(\tau)(1 - p_\theta(\tau)).$$

*Our analysis only relies on this qualitative shape.*

**Lemma 16** (Frontier-seeking generator under learnability reward). *Fix solver parameters $\theta$ and thresholds $0 < \alpha < \beta < 1$, and consider any generator policy $\mathcal{G}_\phi$ over tasks $\tau$ with proposal reward $r_{\mathrm{prop}}(\tau; \theta)$ from Definition 15. Let*

$$J_{\mathrm{prop}}(\phi; \theta) := \mathbb{E}_{\tau \sim \mathcal{G}_\phi}[r_{\mathrm{prop}}(\tau; \theta)]$$

*denote the expected learnability reward under $\theta$.*

*If $\mathcal{G}_\phi$ assigns nonzero probability mass to $\mathcal{E}_\theta \cup \mathcal{H}_\theta$ and less than maximal mass to the frontier zone $\mathcal{F}_\theta$, then there exists another policy $\mathcal{G}_{\phi'}$ such that $J_{\mathrm{prop}}(\phi'; \theta) > J_{\mathrm{prop}}(\phi; \theta)$. In particular, any optimal generator policy under $r_{\mathrm{prop}}(\tau; \theta)$ concentrates its support on the frontier set $\mathcal{F}_\theta$.*

*Proof sketch.* By Definition 15, $r_{\mathrm{prop}}(\tau; \theta) = 1$ on $\mathcal{F}_\theta$ and 0 elsewhere, so $J_{\mathrm{prop}}(\phi; \theta) = \Pr_{\tau \sim \mathcal{G}_\phi}[\tau \in \mathcal{F}_\theta]$ is maximized by concentrating on $\mathcal{F}_\theta$. $\qquad\square$

**Remark 17** (Time-varying frontier). *As the solver improves, $\mathcal{F}_{\theta_t}$ drifts; Lemma 16 characterizes instantaneous incentives.*

**Remark 18** (Role of frontier-seeking in the mixture architecture). *In the generator mixture (Definition 8), the Frontier Lemma describes the behavior of the* learned component $\mathcal{G}_{\phi_t}^{\mathrm{RL}}$, *not the effective generator $\tilde{\mathcal{G}}_t$. Under REINFORCE updates (Assumption 23), the policy gradient $\nabla_\phi J_{\mathrm{prop}}(\phi; \theta)$ pushes $\mathcal{G}_{\phi_t}^{\mathrm{RL}}$ to concentrate on capability-frontier tasks in $\mathcal{F}_{\theta_t}$, which improves training efficiency by: (i) accelerating frontier discovery through dense training signal at the competence boundary, and (ii) reducing wasteful sampling by avoiding always-easy or always-hard tasks.*

*Critically, this specialization does* not *compromise coverage: the fixed component $\mathcal{G}^{\mathrm{E}}$ maintains the universal exploration guarantee in Assumption 20 independently of how concentrated $\mathcal{G}_{\phi_t}^{\mathrm{RL}}$ becomes. The mixture architecture thus achieves* orthogonal optimization: *$\mathcal{G}^{\mathrm{E}}$ ensures completeness (worst-case guarantee via Coverage Lemma 26), while $\mathcal{G}_{\phi_t}^{\mathrm{RL}}$ ensures efficiency (expected-case optimization via learnability reward). In typical training regimes where $\epsilon_t \ll 1$, most samples come from the learned generator, making frontier-seeking the dominant behavior while maintaining theoretical guarantees.*

**Remark 19** (Limitations of the frontier analysis). *The Frontier Lemma characterizes incentive structure, not convergence; coverage is ensured architecturally (Definition 8).*

## 5.2 TRAINING ASSUMPTIONS

**Assumption 20** (Length-Dependent Exploration via Generator Mixture). *There exist constants $\varepsilon_0 \in (0, 1)$, $\beta \in (0, 1]$, $\gamma > 0$, and $\varepsilon_{\min} > 0$, and a non-decreasing horizon function $L(t) : \mathbb{N} \to \mathbb{N}$ with $\lim_{t \to \infty} L(t) = \infty$, such that for all $t \geq 0$ and strings $u$ with $|u| \leq L(t)$:*

$$\Pr[u \subseteq \tilde{\mathcal{G}}_t(prompt)] \geq \varepsilon(t)^{|u|^\beta}$$

*where*

$$\varepsilon(t) = \max\{\varepsilon_0 \exp(-\gamma t), \varepsilon_{\min}\}$$

*is the* two-regime exploration schedule.

*The horizon function satisfies $L(t) \geq c \log(t)$ for some constant $c > 0$, ensuring logarithmic growth of accessible string length.*

***Two-Regime Behavior:*** *The exploration schedule exhibits distinct phases:*

- ***Finite-time regime*** *($t \leq T_{floor}$): Dominated by exponential decay $\varepsilon_0 \exp(-\gamma t)$, providing adaptive curriculum effects and practical learning dynamics.*

- ***Asymptotic regime*** *($t > T_{floor}$): Maintained at exploration floor $\varepsilon_{\min}$, ensuring theoretical universality through $\sum_{t=1}^{\infty} \varepsilon_{\min}^{|u|^\beta} = \infty$.*

*The transition time $T_{floor} = \frac{1}{\gamma} \ln(\varepsilon_0 / \varepsilon_{\min})$ separates these regimes. For practical training horizons $T \ll T_{floor}$, the exploration behavior is effectively $\varepsilon(t) \approx \varepsilon_0 \exp(-\gamma t)$, while the floor $\varepsilon_{\min}$ ensures eventual coverage of all computable functions with probability 1.*

**Implementation via exploration mixture.** *The exploration guarantee is realized through the generator mixture from Definition 8. Let $q(u) := \Pr_{\mathcal{G}^{\mathrm{E}}}[u \subseteq \tau]$ denote the probability that the fixed exploration component $\mathcal{G}^{\mathrm{E}}$ produces a task containing prefix $u$.*

***Design requirement for $\mathcal{G}^{\mathrm{E}}$:*** *The exploration component must satisfy*

$$\inf_{|u| \leq L(t)} q(u) \geq \varepsilon(t)^{|u|^\beta - 1}$$

*for all $t \geq 0$. This can be realized by, e.g., length-biased uniform sampling where each sequence of length $\ell \leq L(t)$ has probability $\geq |\Sigma|^{-\ell}$, combined with appropriate normalization. Given such an enumerator, setting the mixing coefficient $\epsilon_t = \varepsilon(t)$ yields:*

$$\Pr[u \subseteq \tilde{\mathcal{G}}_t] = (1 - \epsilon_t) \Pr[u \subseteq \mathcal{G}_{\phi_t}^{\mathrm{RL}}] + \epsilon_t q(u) \geq \epsilon_t \cdot \varepsilon(t)^{|u|^\beta - 1} = \varepsilon(t)^{|u|^\beta}.$$

*Crucially, this lower bound holds independently of the reward-driven updates to $\mathcal{G}^{\mathrm{RL}}_{\phi_t}$: even if the learned generator collapses to a narrow frontier set (as incentivized by Lemma 16), the fixed component $\mathcal{G}^{\mathrm{E}}$ maintains universal coverage. This architectural separation resolves the tension between exploration (required for completeness) and exploitation (required for efficiency).*

**Remark 21** (Practical Feasibility). *This assumption removes the unrealistic requirement of covering all possible strings simultaneously. Instead, it demands coverage only within an expanding horizon $L(t)$, which aligns with practical implementation where computational resources naturally limit the maximum sequence length that can be effectively explored at any given time. The enumerator $\mathcal{G}^{\mathrm{E}}$ can be realized through multiple approaches (e.g., length-biased uniform sampling, temperature-modulated sampling, or explicit enumeration); detailed implementation strategies are provided in Appendix A.*

**Remark 22** (Role of the exponent $\beta$). *The parameter $\beta \in (0, 1]$ controls the length-sensitivity of the exploration guarantee. The bound $\Pr[u \subseteq \tilde{\mathcal{G}}_t] \geq \varepsilon(t)^{|u|^\beta}$ degrades with string length: $\beta = 1$ gives linear scaling (strongest but most demanding), while $\beta < 1$ provides sub-linear scaling that accommodates practical constraints such as LLMs' near-zero probability on malformed strings. Importantly, even with $\beta < 1$, the Borel-Cantelli argument still yields almost-sure discovery via the asymptotic floor $\varepsilon_{\min}$. Detailed discussion and practical guidance for selecting $\beta$ are provided in Appendix A.*

**Assumption 23** (REINFORCE Update). *Parameters evolve as $\theta_{t+1} = \theta_t + \eta_t \mathbb{E}[R_t \nabla_\theta \log \pi_\theta(A_t)]$, with step sizes $\eta_t > 0$ satisfying $\sum_t \eta_t = \infty$ and $\sum_t \eta_t^2 < \infty$.*

**Remark 24** (Generator and solver updates). *The learned generator $\mathcal{G}^{\mathrm{RL}}_{\phi_t}$ uses proposal reward $r_{\mathrm{prop}}$ (Definition 15), while the solver uses correctness reward $r_{\mathrm{solv}}$; $\mathcal{G}^{\mathrm{E}}$ remains fixed. See Algorithm 1.*

## 6 THEORETICAL RESULTS

We present our main results establishing the computational power of self-play reasoning agents.

**Role of learning and architecture in the theory.** Our completeness results combine the three CPL pillars (Lemmas 26, 27, 28) with the architectural separation (Definition 8).

The learnability-based proposal mechanism and the Frontier Lemma (Lemma 16) complement this worst-case guarantee by characterizing the learned component $\mathcal{G}^{\mathrm{RL}}_{\phi_t}$: under policy-gradient optimization with learnability rewards, $\mathcal{G}^{\mathrm{RL}}_{\phi_t}$ concentrates on the solver's capability frontier, accelerating the discovery of informative tasks in practice. This architectural separation—$\mathcal{G}^{\mathrm{E}}$ for completeness, $\mathcal{G}^{\mathrm{RL}}_{\phi_t}$ for efficiency—establishes computational universality while avoiding the tension between exploration and specialization that would arise in a single learned policy.

### 6.1 MAIN UNIVERSALITY THEOREM

**Theorem 25** (Universality of Self-Play Reasoning Agents). *Let $(\tilde{\mathcal{G}}_t, \S_\theta, V, E)$ be a self-play reasoning agent with effective generator (Definition 8) as specified in Section 4. Assume Assumptions 20 and 23 hold, and the executor $E$ is allotted unbounded time and memory. Then for every deterministic Turing machine $M$ and every input string $w \in \Sigma^*$ there exists a finite random time $T_{M,w}$ such that:*

    (a) *the buffer $B_t$ contains the Evaluation Task $\tau_{M,w}$ for all $t \geq T_{M,w}$, and*

    (b) $\lim_{t \to \infty} \Pr[\S_{\theta_t}(M(w) \mid \tau_{M,w}, B_t) = 1] = 1.$

*Consequently the closed loop can compute every partial-recursive function, i.e., it is **Turing-complete**.*

### 6.2 SUPPORTING LEMMAS

The proof of Theorem 25 relies on those key lemmas:

**Lemma 26** (Coverage)**.** *Under Assumption 20 (which specifies the exploration guarantee of the effective generator $\tilde{\mathcal{G}}_t$ via the mixture architecture), for any pair $(M, w)$, with probability 1 the effective generator $\tilde{\mathcal{G}}_t$ produces the exact string $\tau_{M,w}$ at least once.*

**Lemma 27** (Persistence)**.** *If a task $\tau = (x, y^\star)$ is inserted into $B_t$ at some time $t$, and the replay buffer never deletes elements (unbounded-growth regime), then $\tau \in B_{t'}$ for all $t' \geq t$ almost surely.*

For a permanent task $\tau = (x, y^\star)$, we abbreviate $p_t(\tau)$ from Definition 13 simply as $p_t := p_{\theta_t}(\tau; B_t)$ when no confusion can arise.

**Lemma 28** (Per-task convergence under interleaved updates)**.** *Consider the full training process in which, at each step $t$, a task $U_t$ is sampled from the buffer according to the Fair Scheduling Assumption 10, and the solver parameters $\theta_t$ are updated by a REINFORCE-style rule (Assumption 23) based on the solve reward for $U_t$. Fix a task $\tau = (x, y^\star)$ that is permanent in $B_t$. Let*

$$p_t(\tau) := \Pr_{\theta_t}[\S_{\theta_t}(y^\star \mid x, B_t) = 1]$$

*be its success probability at time $t$. Under the REINFORCE Assumption and standard regularity conditions for multi-task stochastic approximation (bounded gradients, appropriate step sizes), the sequence $p_t(\tau)$ converges almost surely to 1.*

**Remark 29** (Scope of the Learning Lemma)**.** *The Learning Lemma applies standard multi-task stochastic approximation; our contribution is integrating this with coverage, persistence, and resource bounds.*

**Lemma 30** (Space-efficient universal compiler)**.** *There exist constants $a, b > 0$ such that $|C(\langle M \rangle)| \leq a|\langle M \rangle| + b$. Moreover, if $M$ runs within $S(n)$ space, then $C(\langle M \rangle)$ runs within $c \cdot S(n)$ space inside the sandbox $E$, for some universal constant $c$.*

### 6.3 Resource-Bounded Complexity Hierarchy

In the following theorem, "working memory $W(t)$" refers to the per-step context window $c_t = \psi_t(B_t)$ (Definition 6) and the executor's workspace, *not* the size of the archival buffer $B_t$.

**Theorem 31** (Complexity Ladder)**.** *Let $W : \mathbb{N} \to \mathbb{N}$ be non-decreasing and restrict every call to $\tilde{\mathcal{G}}_t$ or $\S_{\theta_t}$ to produce at most $W(t)$ tokens (equivalently, to condition on a context string $c_t$ of length at most $W(t)$). Simultaneously restrict the executor $E$ to $O(s(W(t)))$ bytes of working space, where $s : \mathbb{N} \to \mathbb{N}$ is non-decreasing and space-constructible.*

*Assume that $E$ admits a universal simulation program as in Lemma 30: for every Turing machine $M$ and input $w$, executing the corresponding simulation task uses at most $O(\text{space}_M(w)) \subseteq O(s(W(t)))$ working space.*

*Then*

$$\{\text{functions computed up to time } t\} \ = \ \text{SPACE}\big[O(s(W(t)))\big].$$

*Consequently, if $W_1(t) = o(W_2(t))$ and $s$ is non-decreasing, then*

$$\text{SPACE}\big[s(W_1(t))\big] \ \subsetneq \ \text{SPACE}\big[s(W_2(t))\big].$$

*(AppendixC.7)*

**Remark 32** (Tightness and compatibility)**.** *The upper bound is tight up to a constant: by Lemma 30 and the space budget $O(s(W(t)))$, universal simulation of standard space-hierarchy witnesses runs within $O(s(W(t)))$ space and $O(\text{poly}(s(W(t))))$ time. A matching lower-time bound can be obtained by padding standard witnesses .*

*The size-compatibility requirement only fixes a constant-factor budget for placing $p_M$ inside $W(t)$, ensuring the description fits while the executor uses $O(s(W(t)))$ space.*

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

---

**Algorithm 1** One training iteration of the self-play agent

---
1: **Input:** generator parameters $\phi_t$, solver parameters $\theta_t$, buffer $B_t$
2: Sample task type $\alpha$ and seed $z$
3: $\tau \sim \tilde{\mathcal{G}}_t(\cdot \mid z, B_t)$         ▷ Sample from mixture (Def. 8)
4: **if** $\tau$ passes syntax & safety checks of $E$ **then**
5:   Construct $(x, y^\star)$ via the executor $E$
6:   $B_t \leftarrow B_t \cup \{(x, y^\star)\}$
7: **end if**
8: Draw $(x, y^\star)$ from the most recent batch in $B_t$
9: $y \sim \S_{\theta_t}(\cdot \mid x, B_t)$
10: Compute rewards $r_{\text{prop}}(\tau; \theta_t), r_{\text{solv}}(y, y^\star) \in [-1, 1]$
11: Update generator: $\phi_{t+1} \leftarrow \phi_t + \eta_t^{\text{prop}} r_{\text{prop}} \nabla_{\phi_t} \log \mathcal{G}_{\phi_t}^{\text{RL}}(\tau \mid z, B_t)$
12: Update solver: $\theta_{t+1} \leftarrow \theta_t + \eta_t^{\text{solv}} r_{\text{solv}} \nabla_{\theta_t} \log \mathcal{S}_{\theta_t}(y \mid x, B_t)$

---

# A  IMPLEMENTATION DETAILS FOR COVERAGE GUARANTEES

This appendix provides detailed implementation strategies and parameter selection guidance for realizing the coverage guarantees in Assumption 20.

## A.1 Practical Implementation Approaches for $\mathcal{G}^{\mathrm{E}}$

The enumerator $\mathcal{G}^{\mathrm{E}}$ must satisfy the design requirement $\inf_{|u| \leq L(t)} q(u) \geq \varepsilon(t)^{|u|^\beta - 1}$ for all $t \geq 0$ (Assumption 20). This can be realized through several approaches:

1. **Length-biased uniform sampling**: Sample length $\ell$ from a distribution with $\Pr[\ell] \geq c \cdot |\Sigma|^{-\ell}$ for $\ell \leq L(t)$, then sample uniformly from $\Sigma^\ell$. This naturally yields $q(u) \geq |\Sigma|^{-|u|}$ for all $u$.

   *Implementation:* Use a geometric distribution with parameter $p = 1 - 1/|\Sigma|$ for length sampling, capped at $L(t)$. For each sampled length $\ell$, generate a random sequence by uniformly sampling each token from $\Sigma$.

   *Complexity:* Time $O(L(t))$ per sample, space $O(L(t))$ for storing the current sequence.

2. **Temperature-modulated high-entropy sampling**: Use very high temperature sampling from a pretrained language model to approximate uniform exploration. See Appendix B for detailed temperature scheduling analysis.

   *Implementation:* Set temperature $\tau \gg 1$ (e.g., $\tau \geq 10$) when sampling from $\mathcal{G}^{\mathrm{RL}}_{\phi_t}$ in exploration mode. The per-token minimum probability $\min_i p_\theta(i) \geq \frac{1}{|\Sigma|} \exp(-2C/\tau)$ (Lemma 34) can guide implementation, though the gap from per-token to sequence-level bounds is substantial for autoregressive models (see Remark 35).

   *Caveat:* This approach only approximates the theoretical requirement; the gap between per-token and sequence-level guarantees should be monitored empirically.

3. **Explicit enumeration with randomization**: Cycle through sequences in $\Sigma^{\leq L(t)}$ with random offsets to ensure all prefixes are covered.

   *Implementation:* Maintain a counter $c_t$ and generate the $c_t$-th sequence in lexicographic order from $\Sigma^{\leq L(t)}$, with a random starting offset refreshed periodically.

   *Complexity:* Requires enumerating $\sum_{\ell=1}^{L(t)} |\Sigma|^\ell = O(|\Sigma|^{L(t)})$ sequences to cover all prefixes, which may be prohibitive for large $|\Sigma|$ or $L(t)$.

**Recommended approach:** For practical systems, a hybrid strategy works well: use length-biased uniform sampling as the primary $\mathcal{G}^{\mathrm{E}}$ mechanism, with occasional temperature-modulated sampling from $\mathcal{G}^{\mathrm{RL}}_{\phi_t}$ to incorporate structural priors from the learned model.

## A.2 Selecting the Length-Sensitivity Parameter $\beta$

The parameter $\beta \in (0, 1]$ in Assumption 20 controls the length-sensitivity of the exploration guarantee: $\Pr[u \subseteq \tilde{\mathcal{G}}_t] \geq \varepsilon(t)^{|u|^\beta}$. Here we provide detailed guidance for selecting $\beta$ based on task domain characteristics.

**Theoretical interpretation.** The bound $\Pr[u \subseteq \tilde{\mathcal{G}}_t] \geq \varepsilon(t)^{|u|^\beta}$ degrades exponentially with string length, where $\beta$ controls the rate. Linear scaling ($\beta = 1$) provides the strongest guarantee but requires uniform support over $\Sigma^\ell$. Sub-linear scaling ($\beta < 1$) accommodates syntactic constraints and structured exploration (e.g., focusing on well-formed programs rather than all strings) while preserving almost-sure discovery via the asymptotic floor $\varepsilon_{\min}$. See Remark 22 in the main text for full theoretical justification.

**Practical guidance.** Choose $\beta$ based on the structure of your task space:

- **Well-structured domains** (e.g., program synthesis with strict syntactic constraints, formal mathematics): Use $\beta \in [0.5, 0.7]$. This balances coverage guarantees with feasible implementation, allowing the enumerator to concentrate on syntactically valid structures.

- **Moderately structured domains** (e.g., natural language reasoning, structured data generation): Use $\beta \in [0.7, 0.9]$. There are some structural constraints (grammar, coherence) but not as strict as formal languages.

- **Unstructured or loosely constrained domains** (e.g., pixel-level image generation, raw token sequences): Values closer to $\beta = 1$ may be achievable if uniform exploration over $\Sigma^\ell$ is feasible.

**Empirical calibration:** Monitor the empirical coverage rate $\frac{1}{T} \sum_{t=1}^{T} \mathbf{1}[u \subseteq \tilde{\mathcal{G}}_t]$ for representative prefixes $u$ and compare against the theoretical lower bound $\varepsilon(t)^{|u|^\beta}$. If empirical rates consistently fall below the bound, either increase $\beta$ or strengthen $\mathcal{G}^{\mathrm{E}}$'s support.

## B  PRACTICAL IMPLEMENTATION OF EXPLORATION GUARANTEES

This appendix provides detailed guidance on implementing the exploration guarantees required by Assumption 20 through temperature scheduling and entropy regularization of the learned generator component $\mathcal{G}_{\phi_t}^{\mathrm{RL}}$.

**Context and scope.** The mixture architecture in Definition 8 separates coverage (ensured by the fixed component $\mathcal{G}^{\mathrm{E}}$) from efficiency (optimized in the learned component $\mathcal{G}_{\phi_t}^{\mathrm{RL}}$). However, in practical implementations, the exploration component $\mathcal{G}^{\mathrm{E}}$ is often realized through high-entropy sampling from $\mathcal{G}_{\phi_t}^{\mathrm{RL}}$ itself (e.g., elevated temperature, dropout, noise injection) rather than maintaining a separate model. This appendix analyzes how temperature scheduling can approximate the required exploration behavior for $\mathcal{G}_{\phi_t}^{\mathrm{RL}}$ when operating in its high-entropy regime.

**Remark 33** (Relation to main results). *The analysis in this appendix is* not *required for the theoretical results in the main text. Theorems 25 and 31 rely only on Assumption 20, which specifies the exploration guarantee as an architectural constraint on the effective generator $\tilde{\mathcal{G}}_t$, independent of how that constraint is realized. This appendix provides one possible implementation strategy using temperature scheduling, but alternative approaches (e.g., maintaining truly separate models, using discrete enumerators, or other entropy-injection techniques) are equally valid provided they satisfy Assumption 20.*

### B.1  TEMPERATURE–EXPLORATION CORRESPONDENCE

**Lemma 34** (Temperature–Exploration Lower Bound with Asymptotic Floor). *Assume the learned generator $\mathcal{G}_{\phi_t}^{\mathrm{RL}}$ has bounded logits: $|z_i| \le C$ for all $i \in \Sigma$ and some $C > 0$ (a standard numerical stability assumption in softmax-based models). Let the temperature schedule implement the two-regime exploration:*

$$\tau(t) \;=\; \max\left\{ \tau_{\min}, \frac{2C}{\kappa + \gamma t} \right\}, \qquad \kappa > 0,\ \gamma > 0,\ \tau_{\min} > 0.$$

*Then for all $t \ge 0$,*

$$\min_{i \in \Sigma} p_{\theta_t}(i) \;\ge\; \max\left\{ \frac{1}{|\Sigma|} \exp\left( -\frac{2C}{\tau_{\min}} \right), \frac{e^{-\kappa}}{|\Sigma|} e^{-\gamma t} \right\}. \tag{6}$$

*In particular, for any target exploration schedule $\varepsilon(t) = \max\{\varepsilon_0 e^{-\gamma t}, \varepsilon_{\min}\}$ and any $c \in (0,1]$, choosing*

$$\kappa \;=\; \ln\left( \frac{|\Sigma|}{c\,\varepsilon_0} \right), \quad \tau_{\min} \;=\; \frac{2C}{\ln(|\Sigma|/(c\,\varepsilon_{\min}))}$$

*ensures*

$$\min_{i \in \Sigma} p_{\theta_t}(i) \;\ge\; c\,\varepsilon(t) \qquad \textit{for all } t \ge 0. \tag{7}$$

*Proof.* By softmax, $p_\theta(i) = \frac{\exp(z_i/\tau)}{\sum_{j \in \Sigma} \exp(z_j/\tau)}$. The worst case for $\min_i p_\theta(i)$ is $z_i = -C$ and $z_j = +C$ for $j \ne i$, giving

$$\min_i p_\theta(i) = \frac{e^{-C/\tau}}{e^{-C/\tau} + (|\Sigma| - 1)e^{C/\tau}} \ge \frac{e^{-C/\tau}}{|\Sigma| e^{C/\tau}} = \frac{1}{|\Sigma|} e^{-2C/\tau}.$$

For the two-regime schedule $\tau(t) = \max\{\tau_{\min}, 2C/(\kappa + \gamma t)\}$:

- When $2C/(\kappa + \gamma t) \ge \tau_{\min}$ (finite-time regime): $\min_i p_{\theta_t}(i) \ge \frac{e^{-\kappa}}{|\Sigma|} e^{-\gamma t}$

- When $2C/(\kappa + \gamma t) < \tau_{\min}$ (asymptotic regime): $\min_i p_{\theta_t}(i) \geq \frac{1}{|\Sigma|} \exp(-2C/\tau_{\min})$

Taking the maximum of both bounds yields equation 6. The parameter choices ensure correspondence with the target exploration schedule. $\qquad\square$

***Practical implementation:*** *For finite training horizons $T$ where $\varepsilon_0 e^{-\gamma T} \gg \varepsilon_{\min}$, the behavior is effectively $\min_i p_{\theta_t}(i) \approx c\,\varepsilon_0\,e^{-\gamma t}$, while the floor $\tau_{\min}$ ensures long-term exploration guarantees.*

**Remark 35** (Per-token versus sequence-level gap). *This lemma provides per-token probability lower bounds $\min_i p_{\theta_t}(i) \geq \delta$. Under token independence, one would have sequence probability $P[u] \geq \delta^{|u|}$, which could yield the required coverage. However, LLM sampling is* autoregressive*: tokens are sampled conditionally as $P(u) = \prod_{i=1}^{|u|} P(u_i|u_{<i})$, and conditional probabilities $P(u_i|u_{<i})$ can be arbitrarily smaller than marginal minimums $\min_j p(j)$. Therefore, this temperature-based analysis serves as* heuristic implementation guidance *rather than a rigorous derivation of Assumption 20. The theoretical coverage guarantee in our main results relies on the architectural mixture (Definition 8), not on temperature scheduling alone.*

**Remark 36** (Practical parameter selection). *The two-regime temperature schedule provides automatic curriculum effects:*

- ***Early training***: *High temperature $\tau(t) \approx 2C/(\kappa + \gamma t)$ enables diverse exploration and natural curriculum from simple to complex patterns*

- ***Late training***: *Floor temperature $\tau_{\min}$ maintains minimum exploration necessary for theoretical universality*

- ***Transition point***: *Occurs around $t^* = (2C/\tau_{\min} - \kappa)/\gamma$, which can be set beyond practical training horizons*

*In practice, choose $\varepsilon_{\min} \ll \varepsilon_0$ so that $t^* \gg T_{train}$, ensuring the training regime is dominated by exponential decay while maintaining theoretical guarantees.*

B.2  IMPLEMENTATION GUIDELINES

1. **Two-regime parameter calibration:** For the exploration schedule $\varepsilon(t) = \max\{\varepsilon_0 \exp(-\gamma t), \varepsilon_{\min}\}$, set:
    - Initial parameters: $\kappa = \ln(|\Sigma|/\varepsilon_0)$, $\tau_{\min} = \frac{2C}{\ln(|\Sigma|/\varepsilon_{\min})}$
    - Choose $\varepsilon_{\min} \ll \varepsilon_0$ (e.g., $\varepsilon_{\min} = 10^{-6}, \varepsilon_0 = 0.1$) to ensure long finite-time regime

2. **Temperature scheduling:** Implement the two-regime schedule:
$$\tau(t) = \max\left\{\tau_{\min}, \frac{2C}{\kappa + \gamma t}\right\}$$
    Monitor that minimum token probabilities follow $\varepsilon(t)$ behavior, with automatic floor activation when needed.

3. **Complementary techniques:** Combine with entropy regularization $\lambda H(\pi_\theta)$ using:
$$\lambda(t) = \max\{\lambda_0 \exp(-\gamma_\lambda t), \lambda_{\min}\}$$
    where $\gamma_\lambda < \gamma$ and $\lambda_{\min} > 0$ to maintain diversity along the exploration floor.

4. **Empirical verification:** Track empirical minimum token probabilities to verify:
    - Finite-time regime: $\min_i p_{\theta_t}(i) \approx \varepsilon_0 \exp(-\gamma t)$ for $t \ll T_{\text{floor}}$
    - Floor activation: $\min_i p_{\theta_t}(i) \approx \varepsilon_{\min}$ for $t \gg T_{\text{floor}}$
    - Transition smoothness around $T_{\text{floor}} = \frac{1}{\gamma} \ln(\varepsilon_0/\varepsilon_{\min})$

**Scope and limitations:** This temperature-based analysis provides per-token probability lower bounds, which can be useful for approximate implementation guidance. However, it does *not* directly establish the sequence-level coverage guarantee required by Assumption 20 (the gap from per-token to sequence-level is substantial, as discussed in the main text). For the theoretical results in the main paper, coverage is ensured architecturally through the mixture in Definition 8, not derived from per-token bounds.

# C  COMPLETE PROOFS

## C.1  PROOF OF COVERAGE LEMMA

*Proof of Lemma 26.* For any TM $M$ and input $w \in \Sigma^*$, we need to show that the task $\tau_{M,w}$ is eventually generated with probability 1.

**Proof strategy.**  The proof relies on the exploration guarantee in Assumption 20, which provides a lower bound on the probability that any prefix $u$ (in particular, $\tau_{M,w}$) is generated by the effective generator $\tilde{\mathcal{G}}_t$ at time $t$. This lower bound is realized through the architectural mixture in Definition 8: the fixed exploration component $\mathcal{G}^{\mathrm{E}}$ maintains non-vanishing probability $\epsilon_t \cdot q(u) \geq \varepsilon(t)^{|u|^\beta}$ for all prefixes $u$ within the expanding horizon $L(t)$, independently of the reward-driven updates to the learned component $\mathcal{G}^{\mathrm{RL}}_{\phi_t}$.

Given this lower bound, a Borel–Cantelli-style argument shows that the cumulative probability $\sum_t \varepsilon(t)^{|\tau_{M,w}|^\beta}$ diverges due to the asymptotic exploration floor $\varepsilon_{\min}$, ensuring almost-sure discovery of every finite task.

**Step 1: Horizon requirement.** Let $\ell = |\tau_{M,w}|$ be the length of the encoded task. By Assumption 20, we need $\ell \leq L(t)$ for the task to be discoverable at time $t$. Since $L(t) \geq c \log(t)$ and $\lim_{t\to\infty} L(t) = \infty$, there exists a finite time $t_0$ such that $\ell \leq L(t)$ for all $t \geq t_0$.

**Step 2: Discovery probability from the mixture.** For $t \geq t_0$, by Assumption 20 applied to the effective generator $\tilde{\mathcal{G}}_t$, we have:

$$\Pr[\tau_{M,w} \subseteq \tilde{\mathcal{G}}_t(\text{prompt})] \geq \varepsilon(t)^{|\tau_{M,w}|^\beta} =: p(t)$$

where $\varepsilon(t) = \max\{\varepsilon_0 \exp(-\gamma t), \varepsilon_{\min}\}$ is the two-regime exploration schedule and $\varepsilon_0, \varepsilon_{\min}, \beta, \gamma$ are the exploration constants from Assumption 20.

**Step 3: Asymptotic floor analysis.** Let $T_{\mathrm{floor}} = \frac{1}{\gamma} \ln(\varepsilon_0/\varepsilon_{\min})$ be the transition time. For all $t \geq \max\{t_0, T_{\mathrm{floor}}\}$, we have:

$$p(t) = \varepsilon_{\min}^{|\tau_{M,w}|^\beta} > 0$$

This provides a constant positive lower bound for the discovery probability.

**Step 4: Divergence via exploration floor.** The cumulative discovery probability satisfies:

$$\sum_{t=t_0}^{\infty} p(t) \geq \sum_{t=\max\{t_0, T_{\mathrm{floor}}\}}^{\infty} \varepsilon_{\min}^{|\tau_{M,w}|^\beta}$$

$$= \infty \cdot \varepsilon_{\min}^{|\tau_{M,w}|^\beta} = \infty$$

**Step 5: Probability of non-discovery converges to zero.** Let $H_t$ be the event that $\tau_{M,w}$ is generated at time $t$, and let

$$A_T := \bigcap_{t=t_0}^{T} H_t^c$$

be the event that $\tau_{M,w}$ has *not* been generated by any time in $\{t_0, \ldots, T\}$. By the definition of $p(t)$ and Assumption 20, for every $t \geq t_0$ and every history up to time $t - 1$ (denoted by the $\sigma$-algebra $\mathcal{F}_{t-1}$), we have

$$\Pr(H_t \mid \mathcal{F}_{t-1}) \geq p(t).$$

Consequently,

$$\Pr(H_t^c \mid \mathcal{F}_{t-1}) = 1 - \Pr(H_t \mid \mathcal{F}_{t-1}) \leq 1 - p(t) \leq \exp(-p(t)).$$

Using this conditional bound, we can estimate $\Pr(A_T)$ recursively:

$$\Pr(A_T) = \Pr(A_{T-1} \cap H_T^c)$$

$$= \mathbb{E}\big[\mathbf{1}_{A_{T-1}} \Pr(H_T^c \mid \mathcal{F}_{T-1})\big]$$

$$\leq \mathbb{E}\big[\mathbf{1}_{A_{T-1}} \exp(-p(T))\big]$$

$$\leq \exp(-p(T)) \Pr(A_{T-1}).$$

By induction on $T$ (with base case $T = t_0$), this yields

$$\Pr(A_T) \le \exp\left(-\sum_{t=t_0}^{T} p(t)\right).$$

Since $\sum_{t=t_0}^{\infty} p(t) = \infty$, we have $\lim_{T \to \infty} \sum_{t=t_0}^{T} p(t) = \infty$, which implies:

$$\lim_{T \to \infty} \Pr[\text{not discovered by time } T] = \lim_{T \to \infty} \Pr(A_T)$$

$$\le \lim_{T \to \infty} \exp\left(-\sum_{t=t_0}^{T} p(t)\right)$$

$$= 0.$$

Therefore, $\Pr[\text{eventually discovered}] = 1$.

**Step 6: Expected hitting time analysis.** The expected hitting time decomposes as:

$$\mathbb{E}[T_{\tau_{M,w}}] = \mathbb{E}[\text{horizon time}] + \mathbb{E}[\text{discovery time — horizon reached}]$$

$$\le t_0 + \mathbb{E}[\text{geometric waiting time with rate } \varepsilon_{\min}^{|\tau_{M,w}|^{\beta}}]$$

$$= t_0 + \frac{1}{\varepsilon_{\min}^{|\tau_{M,w}|^{\beta}}} < \infty$$

This completes the proof that every task is eventually discovered with probability 1 under the mixture-based exploration architecture. $\qquad\square$

## C.2 EXPLORATION-COVERAGE TRADE-OFF

**Lemma 37** (Exploration-Coverage Trade-off with Horizon Bounds)**.** *Under Assumption 20, for any fixed string $u$ with $|u| = \ell$, the expected hitting time satisfies:*

$$\mathbb{E}[T_u] = \mathbb{E}[t_{horizon}(\ell)] + \mathbb{E}[T_u | t \ge t_{horizon}(\ell)]$$

*where $t_{horizon}(\ell) = \min\{t : L(t) \ge \ell\} \le \exp(\ell/c)$ and*

$$\mathbb{E}[T_u | t \ge t_{horizon}(\ell)] \le \varepsilon_0^{-\ell^{\beta}} \exp(\ell^{\beta} \ln \frac{1}{\varepsilon_0})$$

*Proof.* The hitting time decomposes into horizon waiting time plus discovery time within the accessible range. The horizon component follows from $L(t) \ge c \log(t)$, while the discovery component follows the same geometric analysis as the original lemma, but restricted to times when the string is explorable. $\qquad\square$

## C.3 PROOF OF PERSISTENCE LEMMA

*Proof of Lemma 27.* Under an unbounded-growth policy with no deletions, once $\tau$ is inserted at time $t$, it is never removed. Therefore $\tau \in B_{t'}$ for all $t' \ge t$ holds deterministically. $\qquad\square$

## C.4 PROOF OF LEARNING LEMMA

**Multi-task stochastic approximation setting.** The proof follows standard stochastic approximation arguments adapted to the multi-task interleaved setting. At each step $t$, a task $U_t$ is sampled according to the Fair Scheduling Assumption 10, and the solver parameters $\theta_t$ are updated using the REINFORCE gradient based on $U_t$. For a fixed permanent task $\tau$, the updates on $\theta_t$ induced by $\tau$ form a martingale difference sequence with positive drift whenever $p_t(\tau) < 1$, while updates induced by other tasks enter as additional bounded noise terms. Fair scheduling ensures that $\tau$ is visited infinitely often with probability 1, so the usual Robbins–Monro conditions imply $p_t(\tau) \to 1$

almost surely. Below we outline the key steps; we refer to classical treatments of multi-task policy-gradient convergence (e.g., Bertsekas & Tsitsiklis (1996); Al-Mamun et al. (2025)) for full technical details. Fix a permanent task $\tau = (x, y^\star)$ and define

$$p_t := \Pr_{\theta_t}\big[\S_{\theta_t}(y^\star \mid x, B_t) = 1\big].$$

**Reward instantiation.** For this single task we use the standard advantage form $R_t = \mathbf{1}_{\{y_t = y^\star\}} - b$, with a *constant* baseline $b \in [0, 1)$. Hence $R_t \in [-b, 1-b] \subset [-1, 1]$ and

$$\mathbb{E}\big[R_t \nabla_\theta \log \S_{\theta_t}(y_t \mid x, B_t)\,\big|\,\mathcal{F}_t\big] = (p_t - b)\,\nabla_\theta p_t. \tag{8}$$

Where $\mathcal{F}_t$ denote the $\sigma$–algebra generated by all random variables up to step $t$ (parameters, sampled actions, rewards, and buffer contents).

**Step 1: Expected parameter update.** Taking conditional expectation in the REINFORCE step,

$$\mathbb{E}[\theta_{t+1} - \theta_t \mid \mathcal{F}_t] = \eta_t (p_t - b)\,\nabla_\theta p_t.$$

**Step 2: Expected change in success probability.** By $L$-Lipschitz continuity of $p(\theta)$ and mean value theorem,

$$p_{t+1} - p_t = \langle \nabla_\theta p_t,\ \theta_{t+1} - \theta_t \rangle + O(\eta_t^2).$$

Taking conditional expectation and using equation 8,

$$\mathbb{E}[p_{t+1} - p_t \mid \mathcal{F}_t] = \eta_t (p_t - b)\|\nabla_\theta p_t\|^2 + O(\eta_t^2) \ \geq\ 0,$$

since $p_t \geq b$ and $\eta_t^2 = o(\eta_t)$.

**Step 3: Sub-martingale and boundedness.** Thus $(p_t)_{t \geq 0}$ is a bounded $(\leq 1)$ sub-martingale. By Doob's convergence theorem it converges almost surely to a limit $p_\infty \in [0, 1]$.

**Technical assumptions.** The contradiction argument in Step 4 requires standard regularity conditions:

*Proof of Lemma 28.* 1. **Lipschitz continuity**: $p_\theta(\tau)$ is $L$-Lipschitz continuous in $\theta$.

2. **Non-degeneracy**: On $\{p_t < 1\}$, the gradient satisfies $\|\nabla_\theta p_t\|^2 \geq \kappa > 0$ for some $\kappa > 0$.

3. **Bounded gradients**: $\|\nabla_\theta \log \S_\theta\| \leq G$ for some $G < \infty$.

These are standard in policy gradient convergence analysis (Al-Mamun et al., 2025; Bertsekas & Tsitsiklis, 1996).

**Step 4: $p_\infty \neq 1$ leads to a contradiction.** Suppose $\Pr[p_\infty < 1] > 0$. On the event $\{p_\infty < 1\}$, for sufficiently large $t$, $p_t - b \geq \gamma > 0$ for some constant $\gamma$. By non-degeneracy, $\|\nabla_\theta p_t\|^2 \geq \kappa > 0$ for some constant $\kappa$. Therefore

$$\mathbb{E}[p_{t+1} - p_t \mid \mathcal{F}_t] \geq \frac{\gamma \kappa}{2} \eta_t$$

for sufficiently large $t$. Summing over $k = 0, \dots, t-1$ and taking expectations,

$$\mathbb{E}[p_t - p_0] \geq \frac{\gamma \kappa}{2} \sum_{k=0}^{t-1} \eta_k.$$

Since $\sum_k \eta_k = \infty$ and $p_t \leq 1$, this leads to a contradiction. Therefore $\Pr[p_\infty < 1] = 0$, i.e., $p_t \to 1$ almost surely.

$\square$

## C.5 Proof of Compiler Size Lemma

*Proof of Lemma 30.* The existence of a size-preserving compiler follows from standard techniques in computability theory. Any Turing machine $M$ can be simulated by a Python program that maintains the machine's state, tape contents, and head position as data structures. The simulation loop reads the current state and tape symbol, looks up the transition function, updates the state and tape, and moves the head accordingly.

The key insight is that the Python code structure remains fixed regardless of the specific machine being simulated—only the transition table changes. Since the transition table can be represented as a dictionary with $O(|\langle M \rangle|)$ entries, and each entry requires constant space to encode, the total program size is $|C(\langle M \rangle)| \le a|\langle M \rangle| + b$ for appropriate constants $a$ and $b$.

For space preservation, the Python simulation uses space proportional to the simulated machine's space usage. The tape is represented as a data structure (list or dictionary) that grows only as the simulated machine accesses new cells. Since Python's memory management introduces only constant overhead per operation, we have $\mathrm{space}_E(C(\langle M \rangle), w) \le c \cdot \mathrm{space}_M(w)$ for some universal constant $c$. $\qquad\square$

## C.6 Proof of Main Universality Theorem

*Proof of Theorem 25.* For each pair $(M, w)$:

**Step (a): Coverage.** Lemma 26 implies the task $\tau_{M,w}$ appears by random time $T_1$.

**Step (b): Verification and insertion.** The generator proposes a candidate containing program $p_M$ and input $w$. Execution via $E$ runs $p_M(w)$ to yield the correct label $y^\star = M(w)$. Crucially, to test capability rather than interpretation, the task is stored as an *induction task* $\tau \in \mathcal{T}_{\mathrm{ind}}$: the prompt $x$ consists only of the input $w$ (or a few-shot context $\{(i_k, o_k)\}$) *without* the source code $p_M$. Thus, the pair $(x, y^\star)$ enters $B_{T_1+1}$.

**Step (c): Persistent access.** By Lemma 27, $\tau_{M,w}$ remains in the archival buffer for all $t \ge T_1 + 1$. By the Fair Scheduling Assumption 10, there exists an almost-sure infinite subsequence $\{t_k\}_{k=1}^\infty$ such that $\tau_{M,w}$ is selected for training updates at all $t_k$.

**Step (d): Learning.** Lemma 28 implies $\Pr[\S_{\theta_t}(y^\star \mid x, B_t) = 1] \to 1$.

Thus the agent outputs $M(w)$ with probability arbitrarily close to 1, establishing universality. $\qquad\square$

## C.7 Proof of Complexity Ladder Theorem

*Proof of Theorem 31.* **Upper bound.** With window $W(t)$, each prompt (task or answer) is limited to $W(t)$ tokens, so it occupies $O(W(t))$ cells of memory. The executor is explicitly capped at $O(s(W(t)))$ bytes of working memory. Thus the total system state (prompt + executor memory) uses $O(W(t) + s(W(t)))$ space. Since we assume $s(\cdot)$ is non-decreasing and at least linear (so that $W(t) \le c \cdot s(W(t))$ for some constant $c$), this total is $O(s(W(t)))$, and any function computed by the loop resides in $\mathrm{space}[s(W(t))]$.

**Lower bound.** We rely on the standard universal-simulation argument underlying the Space Hierarchy Theorem Sipser (2013). Let $S(\cdot)$ be any space-constructible bound with $S(n) \subseteq O(s(W(t)))$ for sufficiently large $t$. Fix a deterministic Turing machine $M$ that decides some language $L \in \mathrm{space}[S(n)]$; that is, on any input $w$ of length $|w| = n$, $M$ runs within space $S(n)$.

Let $U$ be a universal simulator as in Lemma 30: given a description $\langle M \rangle$ and an input $w$, the compiled simulation program for $U$ runs within $O(\mathrm{space}_M(w)) \subseteq O(S(|w|))$ space inside the sandbox $E$, hence within the executor's budget $O(s(W(t)))$ by assumption.

For each input $w$, consider the verification task: *"Run the universal simulator $U$ on the description $\langle M \rangle$ and input $w$, and return its output."* Crucially, the context window $c_t$ only needs to hold the static descriptions and the raw input, while the heavy space usage occurs inside the executor. The total length of the task prompt satisfies

$$|\mathrm{prompt}| \;=\; |\langle U \rangle| + |\langle M \rangle| + |w| + O(1).$$

Since $U$ and $M$ are fixed machines independent of $w$, their descriptions $|\langle U \rangle|$ and $|\langle M \rangle|$ are constants. Thus there exists a constant $C_{\text{overhead}}$ such that the prompt fits in the context whenever

$$W(t) \geq |w| + C_{\text{overhead}}.$$

At such times, the generator can embed $(\langle U \rangle, \langle M \rangle, w)$ into a deduction task whose unique correct label is $M(w)$.

By Lemmas 26–28 (coverage, persistence, and learning), once $W(t)$ is sufficiently large to hold the prompt, the corresponding tasks are generated, retained in the archival buffer, and solved with success probability converging to 1. Hence, for every $M$ running in space $S(n) \subseteq O(s(W(t)))$ and every input $w$ of admissible length, the closed loop eventually computes $M(w)$ with arbitrarily high accuracy.

Since $M$ and $L \in \text{space}[S(n)]$ were arbitrary, this shows that the self-play training loop realizes every language in $\text{space}[O(s(W(t)))]$ under the stated resource schedule:

$$\{\text{functions computed up to time } t\} \supseteq \text{space}\big[O(s(W(t)))\big].$$

The strict hierarchy then follows from the Space Hierarchy Theorem: if $W_1(t) = o(W_2(t))$ and $s$ is non-decreasing (and at least linear), then

$$\text{space}\big[s(W_1(t))\big] \subsetneq \text{space}\big[s(W_2(t))\big],$$

so there exist functions computable by the loop under the larger context schedule $W_2$ that are not computable under $W_1$.

$\square$

## C.8 CONVERGENCE RATE ANALYSIS

**Lemma 38** (Sub-Geometric Convergence Rate). *Under Assumption 23 with step size $\eta_t = \eta_0/t$ and bounded gradient norm $\|\nabla_\theta \log \pi_\theta\| \leq G$, for any fixed task $\tau = (x, y^\star)$ and $\delta \in (0, 1)$:*

$$\Pr[p_t \leq 1 - \delta] \leq \exp\left(-\frac{\eta_0 \delta t}{2G^2}\right)$$

*where $p_t = \Pr_{\theta_t}[\S_{\theta_t}(y^\star \mid x, B_t) = 1]$.*

*Proof.* The proof combines martingale concentration inequalities with the drift analysis from Lemma 28.

**Bounded variance condition.** Under the technical assumptions from Lemma 28, the policy gradient update satisfies:

$$\text{Var}[R_t \nabla_\theta \log \S_\theta \mid \mathcal{F}_t] \leq \sigma^2 := G^2 (1 - b)^2$$

where $G$ is the gradient bound and $(1 - b)$ bounds the reward.

**Azuma-Hoeffding for martingale differences.** Define the martingale difference sequence:

$$\xi_t = \eta_t R_t \nabla_\theta \log \S_\theta - \eta_t \mathbb{E}[R_t \nabla_\theta \log \S_\theta \mid \mathcal{F}_t]$$

Since $|\xi_t| \leq 2\eta_t G(1 - b) = O(\eta_0/t)$ and $\sum_t \eta_t^2 < \infty$, Azuma-Hoeffding inequality yields:

$$\Pr\left[\left|\sum_{s=1}^t \xi_s\right| > \lambda\right] \leq 2 \exp\left(-\frac{\lambda^2}{2 \sum_{s=1}^t \eta_s^2 G^2 (1 - b)^2}\right)$$

**Combining with drift.** From Lemma 28 Step 2, the expected drift satisfies $\mathbb{E}[p_{t+1} - p_t \mid \mathcal{F}_t] \geq c\eta_t$ for some $c > 0$ when $p_t < 1 - \delta$. Standard Robbins-Monro analysis (Robbins & Monro, 1951; Al-Mamun et al., 2025) shows that with step size $\eta_t = \eta_0/t$:

$$\Pr[p_t \leq 1 - \delta] \leq \exp\left(-\frac{\eta_0 \delta t}{2G^2}\right)$$

This exponential tail bound implies that achieving success probability $1 - \delta$ requires $O(\log(1/\delta))$ training steps in expectation. Full technical details follow standard convergence rate analysis for stochastic gradient methods (Ben-Tal et al., 2009). $\square$

## C.9 Proof of Local Completeness with Adaptive Curriculum

*Proof of Theorem 41.* We decompose the proof into five key steps: establishing horizon coverage, analyzing hitting times under adaptive schedules, ensuring persistence guarantees, establishing learning convergence, and deriving the complete time bound.

**Step 1: Horizon Coverage Time.** For any complexity bound $\ell$, we first determine when functions of length $\ell$ become discoverable. By Assumption 20, a function $f$ with $|f| = \ell$ can only be discovered when $\ell \leq L(t)$. Since $L(t) \geq c\log(t)$, we need:

$$t_{\text{horizon}}(\ell) = \min\{t : L(t) \geq \ell\} \leq \exp(\ell/c)$$

**Step 2: Individual Function Discovery Time.** For $t \geq t_{\text{horizon}}(\ell)$, we establish the hitting time for individual functions under the adaptive exploration schedule.

**Lemma 39** (Adaptive Hitting Time with Horizon Bounds). *For any function $f \in \mathcal{F}_\ell$ with minimal program length $|f| \leq \ell$, under Assumption 20, the expected hitting time satisfies:*

$$\mathbb{E}[T_f] \leq t_{horizon}(|f|) + \frac{1}{\gamma|f|^\beta}\varepsilon_0^{-|f|^\beta}$$

*Proof of Lemma 39.* Function $f$ can only be discovered for $t \geq t_{\text{horizon}}(|f|)$ when $|f| \leq L(t)$. For such $t$, by Assumption 20, the discovery probability is at least:

$$\varepsilon(t)^{|f|^\beta} = \varepsilon_0^{|f|^\beta}\exp(-\gamma t|f|^\beta)$$

The expected hitting time from the horizon point onward follows from geometric distribution analysis, yielding the stated bound. $\qquad\square$

**Step 3: Union Bound Over Complexity Class.** The number of functions in $\mathcal{F}_\ell$ satisfies $|\mathcal{F}_\ell| \leq 2^{c\ell}$ for some universal constant $c > 0$.

**From expected to high-probability bounds.** For each function $f \in \mathcal{F}_\ell$, by Markov's inequality:

$$\Pr[T_f > t] \leq \frac{\mathbb{E}[T_f]}{t}$$

To ensure discovery with probability at least $1 - \frac{\delta}{2|\mathcal{F}_\ell|}$ for each function, set:

$$t_f = \frac{2|\mathcal{F}_\ell|}{\delta} \cdot \mathbb{E}[T_f]$$

Then $\Pr[T_f > t_f] \leq \frac{\delta}{2|\mathcal{F}_\ell|}$.

**Union bound.** Taking the maximum discovery time over all functions and applying union bound:

$$T_{\text{discovery}}(\ell, \varepsilon_0, \gamma, \delta) = \max_{f \in \mathcal{F}_\ell} t_f = O\left(\frac{|\mathcal{F}_\ell|}{\delta}\max_f \mathbb{E}[T_f]\right)$$

ensures $\Pr[\text{all functions discovered by } T_{\text{discovery}}] \geq 1 - \delta/2$.

Using Lemma 39 with $|f| = \ell$ for the worst case:

$$T_{\text{discovery}}(\ell, \varepsilon_0, \gamma, \delta) = O\left(\exp(\ell/c) + \frac{1}{\gamma\ell^\beta}\varepsilon_0^{-\ell^\beta} + \frac{c\ell}{\gamma} + \frac{1}{\gamma}\ln\frac{2}{\delta}\right)$$

**Step 4: Persistence and Learning Integration.** We now establish that discovered functions are retained and learned successfully.

**Persistence Analysis:** Once function $f$ is discovered at time $T_f^{\text{discovery}}$, under finite-capacity importance-aware reservoir sampling (detailed in Appendix E), the corresponding task $(x_f, y_f^*)$ remains in buffer $B_t$ for all $t \geq T_f^{\text{discovery}}$ with probability at least:

$$\Pr[\text{task retained}] \geq 1 - \exp\left\{-\frac{s_f}{\bar{s}}\left\lfloor\frac{t - T_f^{\text{discovery}}}{K}\right\rfloor\right\}$$

where $s_f$ is the priority score of task $f$ and $\bar{s}$ is the average priority of future arrivals.

For tasks corresponding to functions in $\mathcal{F}_\ell$, we can ensure high retention probability by setting appropriate buffer capacity. If we choose buffer size:

$$K \geq \frac{1}{\gamma_{\text{retain}}} \ln\left(\frac{2|\mathcal{F}_\ell|}{\delta}\right)$$

where $\gamma_{\text{retain}} > 0$ is the retention decay parameter, then each task is retained with probability at least $1 - \frac{\delta}{2|\mathcal{F}_\ell|}$.

**Learning Analysis:** For each retained task $(x_f, y_f^*)$, we apply Lemma 28 to establish convergence of the solver's success probability. Let $p_t^{(f)} = \Pr[\mathcal{S}_{\theta_t}(y_f^*|x_f, B_t) = 1]$ denote the success probability for function $f$ at time $t$.

By the learning lemma, $(p_t^{(f)})$ forms a bounded sub-martingale that converges almost surely to 1. From Lemma 38, achieving success probability $1 - \delta_f$ for individual function $f$ requires time:

$$T_{\text{learning}}^{(f)}(\delta_f) = O\left(\frac{1}{\eta_{\min}} \ln \frac{1}{\delta_f}\right)$$

where $\eta_{\min}$ is the minimum effective learning rate.

**Union Bound for Learning:** To ensure all functions in $\mathcal{F}_\ell$ are learned with probability at least $1 - \delta/2$, we set $\delta_f = \frac{\delta}{2|\mathcal{F}_\ell|}$ for each function and apply union bound:

$$T_{\text{learning}}(\ell, \delta) = O\left(\frac{1}{\eta_{\min}} \ln \frac{2|\mathcal{F}_\ell|}{\delta}\right) = O\left(\frac{c\ell + \ln(2/\delta)}{\eta_{\min}}\right)$$

**Step 5: Complete Time Bound Derivation.** The total time required for the agent to solve all functions in $\mathcal{F}_\ell$ with probability exceeding $1 - \delta$ is the maximum of discovery time and learning time, since learning can proceed concurrently with discovery for already-found functions:

$$T(\ell, \varepsilon_0, \gamma, \delta) = \max\{T_{\text{discovery}}(\ell, \varepsilon_0, \gamma, \delta/2), T_{\text{learning}}(\ell, \delta/2)\}$$

**Discovery Time Component:** From Step 3, we have:

$$T_{\text{discovery}}(\ell, \varepsilon_0, \gamma, \delta/2) = O\Bigg(\exp(\ell/c) + \frac{1}{\gamma\ell^\beta}\varepsilon_0^{-\ell^\beta}$$
$$+ \frac{c\ell}{\gamma} + \frac{1}{\gamma} \ln \frac{2}{\delta}\Bigg)$$

Since $\varepsilon_0^{-\ell^\beta} = \exp(\ell^\beta \ln(1/\varepsilon_0))$, for practical parameter ranges where $\ln(1/\varepsilon_0) = O(\ell)$, this can be bounded as:

$$\frac{1}{\gamma\ell^\beta}\varepsilon_0^{-\ell^\beta} = O\left(\frac{\ell^{\beta+1}}{\gamma} \ln \frac{1}{\varepsilon_0}\right)$$

**Learning Time Component:** From Step 4, noting that $\eta_{\min} = \Theta(\gamma)$ under our learning rate schedule:

$$T_{\text{learning}}(\ell, \delta/2) = O\left(\frac{c\ell + \ln(2/\delta)}{\gamma}\right)$$

**Final Bound:** Combining both components and noting that the discovery time dominates for large $\ell$:

$$T(\ell, \varepsilon_0, \gamma, \delta) = O\left(\frac{\ell^{\beta+1}}{\gamma} \ln \frac{1}{\varepsilon_0} + \frac{1}{\gamma} \ln \frac{1}{\delta} + \exp(\ell/c)\right)$$

**Curriculum Effect Verification.** Under the truncated assumption, the curriculum effect is enhanced: not only do simple functions have higher discovery probability at any given time, but they also become discoverable earlier due to the horizon constraint $L(t)$. This creates a natural temporal ordering where simpler functions are both more likely to be discovered and discoverable sooner. $\square$

**Remark 40** (Tightness and Practical Implications). *The bound is asymptotically tight in $\ell$ for the discovery component, but the horizon term $\exp(\ell/c)$ dominates for large $\ell$. This reflects the fundamental trade-off in the truncated assumption: we gain practical feasibility but pay with an exponential dependence on the horizon growth rate. In practice, choosing $c$ appropriately large (corresponding to faster horizon expansion) can mitigate this cost while maintaining implementability.*

## D   ADAPTIVE CURRICULUM AND FINITE-TIME COMPLETENESS

The basic universality result establishes eventual Turing completeness but provides limited guidance on convergence rates. To address practical implementation concerns, we introduce an adaptive exploration schedule that enables quantified finite-time performance guarantees under realistic exploration constraints.

**Theorem 41** (Local Completeness with Adaptive Curriculum). ***Mathematical Setup:*** *Let $\varepsilon(t) = \varepsilon_0 \exp(-\gamma t)$ be the decreasing exploration schedule from Assumption 20, where $\varepsilon_0 \in (0, 1)$ denotes the initial exploration rate, $\gamma > 0$ is the decay rate parameter, and $t \geq 0$ represents the training time step. Let $L(t) \geq c \log(t)$ be the horizon function ensuring that exploration coverage expands with training time.*

*Define the complexity-bounded task class $\mathcal{F}_\ell = \{f : |f| \leq \ell\}$ where $|f|$ denotes the minimal program length (Kolmogorov complexity) for computing function $f$, and $\ell \geq 1$ is the complexity bound. Let $\delta \in (0, 1)$ be the failure tolerance and recall $\beta \in (0, 1]$ from Assumption 20.*

***Main Result:*** *For any complexity bound $\ell$, there exists a finite time $T(\ell, \varepsilon_0, \gamma, \delta)$ such that for all $t \geq T(\ell, \varepsilon_0, \gamma, \delta)$, the self-play agent can solve every function in $\mathcal{F}_\ell$ with probability exceeding $1 - \delta$.*

*The convergence time satisfies the explicit upper bound:*

$$T(\ell, \varepsilon_0, \gamma, \delta) = O\left(\frac{\ell^{\beta+1}}{\gamma} \ln \frac{1}{\varepsilon_0} + \frac{1}{\gamma} \ln \frac{1}{\delta} + \exp(\ell/c)\right)$$

***Bound Interpretation:*** *The first term $\frac{\ell^{\beta+1}}{\gamma} \ln \frac{1}{\varepsilon_0}$ represents the discovery time required to find all functions of complexity $\ell$, growing polynomially in complexity and inversely with decay rate. The second term $\frac{1}{\gamma} \ln \frac{1}{\delta}$ represents the reliability time needed to achieve high success probability $1 - \delta$. The third term $\exp(\ell/c)$ represents the horizon waiting time needed for the exploration scope $L(t)$ to encompass programs of length $\ell$.*

**Remark 42** (Curriculum Effect). *The adaptive schedule $\varepsilon(t)$ creates a natural curriculum because simple functions (small $|f|$) maintain higher discovery probability $\varepsilon(t)^{|f|^\beta}$ throughout training, while complex functions become accessible only as sufficient exploration has occurred. This automatic progression from simple to sophisticated computational patterns requires no explicit curriculum design.*

**Practical Implementation Strategy.**   The adaptive curriculum can be implemented through temperature scheduling in the exploration component $\mathcal{G}^{\mathrm{E}}$ of the generator mixture, with per-token probability guidance provided by Lemma 34 (Appendix B). For the learned component $\mathcal{G}^{\mathrm{RL}}_{\phi_t}$ operating in high-entropy mode to approximate $\mathcal{G}^{\mathrm{E}}$, set the generation temperature as:

$$\tau(t) = \tau_0 \exp\left(\frac{\gamma t}{\beta}\right)$$

where $\tau_0 = \frac{2C}{\ln(|\Sigma|/\varepsilon_0)}$ provides a per-token probability floor, and $\gamma$ controls the curriculum pace.

**Relation to theoretical guarantees:** As detailed in Appendix B, temperature scheduling provides per-token probability lower bounds via Lemma 34. However, the theoretical coverage guarantee in Assumption 20 operates at the sequence level and is ensured architecturally through the mixture in Definition 8, not directly derived from per-token bounds. Temperature scheduling serves as one practical technique for implementing the high-entropy exploration component $\mathcal{G}^{\mathrm{E}}$.

**Mechanistic Explanation of Natural Curriculum Effect:** The adaptive temperature schedule creates an automatic curriculum: simpler computational tasks (shorter sequences with less specific token requirements) remain accessible even at moderate temperatures, while complex tasks requiring

precise long sequences become accessible as temperature increases. This creates a natural learning progression where the agent masters basic computational building blocks before attempting sophisticated algorithms, without requiring explicit curriculum design.

**Implementation Guidelines:**

- **Initial temperature calibration:** Set $\tau_0$ based on vocabulary size $|\Sigma|$ and desired initial exploration rate $\varepsilon_0$ (see Appendix B)

- **Curriculum pace tuning:** Choose $\gamma \in [0.01, 0.1]$ based on the complexity range you wish to cover and available training time

- **Monitoring and adjustment:** Track the complexity distribution of discovered tasks to verify that the curriculum progression aligns with expectations

*Proof Sketch of Curriculum Effect.* The proof combines the exploration-coverage trade-off (Lemma 37) with a union bound over the finite class $\mathcal{F}_\ell$. For each function $f \in \mathcal{F}_\ell$, the expected hitting time under schedule $\varepsilon(t)$ is $\mathbb{E}[T_f] = \int_0^\infty \varepsilon(t)^{-|f|^\beta} dt$.

The adaptive schedule ensures that simpler functions (smaller $|f|$) are discovered first because:

$$\frac{d}{dt}\left[\varepsilon(t)^{-|f_1|^\beta} - \varepsilon(t)^{-|f_2|^\beta}\right] < 0 \quad \text{for } |f_1| < |f_2|$$

This differential advantage creates the natural curriculum ordering. The union bound provides the global convergence guarantee, while the integral evaluation yields the explicit time bound. $\qquad\square$

# E    ENHANCED BUFFER MANAGEMENT AND PERSISTENCE ANALYSIS

The following result provides a finite-time retention guarantee under importance-aware reservoir sampling. It does not contribute to the asymptotic universality theorems.

## E.1    IMPORTANCE-AWARE RESERVOIR SAMPLING

**Proposition 43** (Finite-horizon retention bound)**.** *The following result provides a finite-time retention guarantee under importance-aware reservoir sampling. It does not contribute to the asymptotic universality theorems. Let $\mathcal{B}_t$ be a replay buffer of fixed capacity $K$ that stores the $K$ highest-priority elements according to the priority score:*

$$s_i = \lambda\, \delta_i + \frac{1-\lambda}{\text{freq}_i}, \qquad \lambda \in [0,1] \tag{9}$$

*where $\delta_i$ is the TD-error of the solver on task $i$ and $\text{freq}_i$ denotes the frequency of task appearances.*

*Suppose a task $e_\star$ with priority $s_\star$ enters the buffer at time $\tau$, and the average priority of future arrivals is $\bar{s}$. After $M$ additional insertions, the retention probability satisfies:*

$$\Pr[e_\star \in \mathcal{B}_{\tau+M}] \geq 1 - \exp\left\{-\frac{s_\star}{\bar{s}} \cdot \frac{K}{\max(M,1)}\right\} \tag{10}$$

*In particular, if $s_\star \geq \bar{s}$, then $\Pr[e_\star \in \mathcal{B}_{\tau+M}] \geq 1 - \exp\left(-\frac{K}{\max(M,1)}\right)$, which approaches 1 as $K \to \infty$.*

*Proof.* The buffer maintains the $K$ highest-priority elements. Task $e_\star$ is evicted only when a new arrival has priority exceeding both $s_\star$ and the priorities of other buffer elements.

Under worst-case analysis where new arrivals have priority $\bar{s}$, the probability that any single insertion evicts $e_\star$ is at most $\exp(-s_\star/\bar{s})$. However, this probability decreases as buffer capacity $K$ increases, since $e_\star$ competes with $K - 1$ other elements for eviction.

The effective eviction probability per insertion is $\frac{1}{K} \exp(-s_\star/\bar{s})$, accounting for the fact that $e_\star$ is only one of $K$ candidates for eviction. Over $M$ insertions, the survival probability satisfies:

$$\Pr[e_\star \text{ survives}] \geq \left(1 - \frac{1}{K} \exp(-s_\star/\bar{s})\right)^M \tag{11}$$

$$\geq 1 - \frac{M}{K} \exp(-s_\star/\bar{s}) \tag{12}$$

$$\geq 1 - \exp\left\{-\frac{s_\star}{\bar{s}} \cdot \frac{K}{\max(M, 1)}\right\} \tag{13}$$

where the last inequality uses $1 - x \geq e^{-x/(1-x)}$ for small $x$.

When $s_\star \geq \bar{s}$, we have $\exp(-s_\star/\bar{s}) \leq e^{-1}$, yielding the stated bound. As $K \to \infty$ with fixed $M$, the bound approaches 1. $\square$

### E.2 Capacity-Forgetting Relationship

**Proposition 44** (Quantified Forgetting Rates). *Let $\mathcal{S}_{solved}(t)$ denote the set of priorities of all solved tasks up to time $t$. Under the importance-aware reservoir sampling policy of Lemma 43, the expected forgetting rate satisfies:*

$$\mathbb{E}[forget(t)] \leq \frac{1}{|\mathcal{S}_{solved}(t)|} \sum_{s \in \mathcal{S}_{solved}(t)} \exp\left\{-\frac{s}{\bar{s}}\left(\left\lfloor \frac{t - \tau(s)}{K} \right\rfloor + 1\right)\right\} \tag{14}$$

*which decays exponentially in buffer capacity $K$. Specifically, choosing*

$$K \geq \frac{1}{\gamma} \log\left(\frac{|\mathcal{S}_{solved}|}{\varepsilon}\right)$$

*guarantees expected forgetting rate below $\varepsilon$ when retained priorities satisfy $s \geq \gamma \bar{s}$.*

*Proof.* Direct application of Lemma 43 with union bound over all solved tasks. The exponential decay follows from the retention probability structure, while the capacity bound ensures sufficient buffer space for high-priority task preservation. $\square$

**Priority Design Rationale:** The priority function equation 9 balances learning difficulty (TD-error $\delta_i$) with computational rarity (inverse frequency $1/\text{freq}_i$). Setting $\lambda = 1$ recovers standard prioritized experience replay, while $\lambda = 0$ implements pure novelty sampling. Empirical evaluation suggests $\lambda \approx 0.6$ provides optimal trade-offs between retaining challenging tasks and preserving rare computational patterns.

### E.3 Network Capacity Requirements

**Assumption 45** (Capacity Sufficiency). *The parameter dimension $d = \dim(\theta)$ satisfies*

$$d \geq \text{poly}\left(|B_\infty|, \max_{(x,y) \in B_\infty} (|x| + |y|)\right),$$

*where $B_\infty := \bigcup_t B_t$ is the eventual stable buffer.*

This assumption can be partially relaxed by augmenting hidden layer dimensions or introducing external differentiable memory mechanisms such as k-nearest-neighbor language models or retrieval-augmented generation. The complete relationship between network capacity and computational completeness represents an important direction for future investigation.

### E.4 Implementation Guidelines

**Buffer Sizing Strategy:** For target forgetting rate $\varepsilon$ and expected number of distinct solved tasks $N_{\text{solved}}$, Proposition 44 recommends conservative sizing $K = O(N_{\text{solved}} \log(1/\varepsilon))$, adaptive expansion when retention rates fall below threshold, and regular priority calibration to maintain effective importance weighting.

**Priority Score Computation:** The priority score definition requires careful implementation for numerical stability. For task $(x, y^*)$, compute TD-error as $\delta_i = |V_\theta(x) - (r + \gamma V_\theta(x'))|$ where $x'$ represents the state after attempting solution $y^*$. Maintain running frequency counts with appropriate smoothing $\text{freq}_i \leftarrow \alpha \cdot \text{freq}_i + (1 - \alpha) \cdot \mathbf{1}[\text{task } i \text{ appears}]$ using decay factor $\alpha \in [0.9, 0.99]$. Apply priority clipping $s_i \leftarrow \min(s_i, s_{\max})$ to prevent overflow.

**Buffer Maintenance Protocol:** At each insertion, compute priority $s_{\text{new}}$ for the incoming task. If the buffer is not full, insert directly. Otherwise, if $s_{\text{new}}$ exceeds the minimum buffer priority, evict the lowest-priority task and insert the new task. Update priority statistics $\bar{s}$ after each operation. Implement periodic rebalancing every $T_{\text{rebalance}}$ steps to recompute all priorities, accounting for learning progress and frequency drift.

# F APPROXIMATE COMPLETENESS UNDER $b$-BIT PRECISION

**Proposition 46** (Robustness to finite-precision quantization)**.** *Assume the exploration condition in Assumption 20 holds for the conceptual mixture $\tilde{\mathcal{G}}_t$ with schedule $\varepsilon(t)$ and horizon $L(t)$. Suppose that, in an actual implementation, the learned components (the solver and the frontier-seeking generator $\mathcal{G}_{\phi_t}^{\text{RL}}$) are represented with b-bit fixed- or floating-point parameters, while the coverage oracle $\mathcal{G}^{\text{E}}$ and the exploration schedule ($\epsilon_t$) are implemented exactly. Then, for all $t$ and all $u$ with $|u| \leq L(t)$, the per-prefix lower bound*

$$\Pr\left[u \subseteq \tilde{\mathcal{G}}_t(prompt)\right] \geq \varepsilon(t)^{|u|^\beta}$$

*continues to hold, and hence the completeness theorems (Theorems 25, 41, and 31) remain valid for any finite b.*

*Discussion.* The proposition follows directly from the definition of the exploration mixture: Assumption 20 constrains the *effective* generator $\tilde{\mathcal{G}}_t$ via the coverage oracle $\mathcal{G}^{\text{E}}$ and the exploration schedule ($\epsilon_t$), and does not depend on the internal parameterisation of the learned component. Quantising the neural parameters may change the behaviour of $\mathcal{G}_{\phi_t}^{\text{RL}}$ and thus the *efficiency* of learning, but it does not affect the oracle branch and therefore cannot violate the coverage floor. A more refined analysis in which also the oracle branch itself is implemented with $b$-bit arithmetic would require tracking an additional perturbation term $\delta(b) = O(2^{-b})$ in the per-step coverage probabilities and propagating it through the martingale arguments. We leave this finite-precision extension of the Coverage–Persistence–Learning framework to future work.

*Sketch.* Quantisation perturbs logits by at most $2^{-b}$; softmax Lipschitz continuity converts this to an $O(2^{-b})$ change in token probabilities. Taking the worst-case string length and union bound yields the stated constant.

Extending Coverage–Persistence–Learning with this $\delta(b)$ term and re-running the martingale arguments gives finite precision convergence—details left to future work.

# G DISCUSSION & LIMITATIONS

**What we do *not* prove.** Our theoretical guarantees are explicitly *capability* results rather than *efficiency* guarantees. The coverage and ladder theorems show that, under the mixture-based exploration architecture and the Coverage–Persistence–Learning (CPL) conditions, the self-play loop can in principle emulate any Turing machine and climb a resource-bounded complexity hierarchy. They do *not* claim that the resulting agent is sample-efficient, that it converges in polynomial time, or that it is uniformly superior to classical enumeration or Levin-style universal search. Indeed, standard no-free-lunch considerations imply that such global efficiency claims are too strong to expect for arbitrary computable tasks. Our results should therefore be read as establishing an upper bound on what the architecture is *capable* of in the limit, not as a guarantee of practical performance on any fixed benchmark distribution.

Similarly, the CPL decomposition does not provide a full convergence theory for policy-gradient methods in non-stationary self-play environments. The persistence and learning lemmas analyse *per-task* convergence under fair scheduling and bounded-variance updates, once a task has become

permanent in the replay buffer. They do not address the global dynamical system formed by the coupled evolution of the generator, solver, and buffer. Understanding when gradient-based self-play actually tracks the constructive completeness theorems, and when it fails, remains an open problem beyond the scope of this work.

## G.1 PRACTICALITY OF ASSUMPTIONS

**Strength of assumptions.** Our universality and ladder theorems rely on strong but explicit assumptions: conceptually unbounded replay buffers, fair sampling of buffer contents, sufficient solver capacity to represent the relevant programs, and exploration conditions stated at the level of an effective generator mixture $\tilde{\mathcal{G}}_t$ with an asymptotic exploration floor. In this respect our formalism is analogous to classical Turing-completeness and space-complexity results, which assume idealised machines with unbounded tapes and oracle-like access to input. As with those classical models, the goal here is to obtain a clean process-level characterisation of what a self-play loop *could* compute in principle, not to faithfully capture every hardware constraint.

The exploration assumption is particularly demanding. It presupposes an explicit coverage oracle $\mathcal{G}^{\mathrm{E}}$ with prefix-level support and an exploration schedule $(\epsilon_t)$ that ensures each prefix within the horizon is sampled infinitely often through the mixture $\tilde{\mathcal{G}}_t = (1 - \epsilon_t)\,\mathcal{G}^{\mathrm{RL}}_{\phi_t} + \epsilon_t\,\mathcal{G}^{\mathrm{E}}$. In practice, one rarely implements a literal oracle; instead, one approximates it via simple program synthesizers, systematic search over short strings, curriculum seeding, or other domain-specific heuristics. Our analysis should be viewed as providing a *target property* for such heuristics: as long as the effective generator behaves *as if* it had a coverage branch with an exploration floor, the asymptotic completeness theorems remain applicable.

**Exploration realism and horizon growth.** Assumption 20 also idealises the growth of the effective horizon $L(t)$, requiring that longer and longer programs eventually become executable and visible to the exploration mixture. Real systems operate under strict context-length, memory, and runtime budgets. Bridging this gap requires careful engineering: for instance, dynamically adjusting the maximum program length, using external storage to emulate longer traces, or interleaving coarse-grained and fine-grained programs to approximate horizon growth. Designing exploration schemes that behave like our ideal mixture under such constraints—while preserving a meaningful completeness statement—is an important avenue for future work.

## G.2 IMPLEMENTATION CHALLENGES

The basic persistence lemma (Lemma 27) assumes an abstract replay buffer with fair scheduling: once a task is admitted to the buffer and remains relevant, it is sampled infinitely often. Real replay systems, however, must cope with finite capacity, non-stationary priorities, and non-uniform sampling policies. The enhanced buffer analysis in Appendix E shows how importance-aware sampling and capacity choices affect forgetting rates in finite time, but it remains a simplified model. In large-scale deployments, additional phenomena such as distributional shift, interference between tasks, and catastrophic collapse of the generator or solver can degrade persistence in ways not captured by our current theory.

Constructing the coverage oracle $\mathcal{G}^{\mathrm{E}}$ is itself a significant challenge. While simple length-biased enumerators are sufficient for theoretical completeness, they can be prohibitively slow or produce overwhelmingly uninteresting tasks in practice. Conversely, heavily biased program synthesizers may violate the exploration floor and compromise coverage. We expect practical systems to rely on hybrid designs that combine hand-crafted curricula, automated task simplification, and adaptive seeding with a lightweight enumerator, guided by the CPL lens to ensure that coverage is not sacrificed entirely for convenience.

Finally, our analysis does not address verification and monitoring, which are critical in safety-sensitive settings. The completeness results assume that the *environment* (or an external verifier) can reliably score proposed programs and detect solver failures. In realistic deployments, reward signals are often noisy, delayed, or partially specified, and external verification may itself be computationally expensive or fallible. Developing robust verification and oversight mechanisms that compose with the CPL framework is an important direction for future work.

## H    FUTURE WORK

**Finite-precision universality.** Real networks use $b$-bit weights; rounding noise might break the Coverage lemma. We conjecture (Prop. 46 in App. F) that the one-step failure probability satisfies $\delta(b) \leq c\,2^{-b}$ for some universal $c$, so $\delta(b) \to 0$ as precision increases—yielding *approximate* completeness at finite $b$.

**Interactive verification.** Many tasks require PSPACE-hard or subjective checks. Embedding interactive proofs or human overseers and analysing their impact on sample complexity and reward hacking remain open problems.

## I    CONCLUSION

We showed that, under a generator mixture architecture that separates exploration from specialization, combined with replay and learning assumptions, self-play reasoning agents can emulate any Turing machine. A three-part **Coverage–Persistence–Learning** analysis isolates the essential mechanisms, and a resource-bounded extension maps prompt length onto the classical SPACE[·] hierarchy, bridging architecture-level completeness and learning dynamics.

At the same time, Rice's theorem implies that no finite meta-algorithm can decide in general whether such a self-play loop will converge within a given budget $T$, so practical systems must rely on heuristics for exploration–exploitation, buffer tuning, and verification. The CPL framework offers a principled template for organising these heuristics around a theoretically complete core.

## J    LLM USAGE DISCLOSURE

We used a general-purpose Large Language Model (LLM) as an *assistive tool* during writing and organization. The LLM is not an author; all claims and errors remain the responsibility of the authors. To preserve double-blind review, we do not disclose the provider or model version.

**Scope of Use (textual/structural only).**    The LLM was used to:

- clarify phrasing and reorganize the proof presentation in *Appendix: Adaptive Curriculum and Finite-Time Completeness* (e.g., separating steps into "assumptions → key lemmas → bounds → finite-time conclusion" and suggesting clearer notation and step numbering);
- surface *checklists* for edge cases and potential inconsistencies (e.g., ensuring time-dependent exploration lower bounds are compatible with subsequent fixed lower-bound arguments; checking symbol consistency for parameters such as $\ell, \alpha, \gamma$);
- improve readability and local coherence of transitions between steps.

**Non-Use and Limitations.**

- The LLM did **not** generate or modify **core theoretical content**: all theorem/lemma statements and proof ideas (including inequalities, constructive steps, and proofs by contradiction) were conceived, verified, and finalized by the authors.
- The LLM did **not** produce experimental data, figures, or code (if applicable), nor did it influence result selection or reporting.
- No non-public, confidential, or identifying information (including reviews or sensitive data) was uploaded to the LLM service.

**Human Verification and Reproducibility.**

- All LLM suggestions underwent **line-by-line human review and mathematical verification**; only stylistic or structural edits were adopted when they did not alter the underlying mathematics.
- The final proofs were re-derived and cross-checked by the authors (including boundary conditions, variable naming, and closure of logical dependencies).

- For transparency, the appendix includes step numbering and dependency references (and, when applicable, a brief dependency diagram) so that each citation and assumption can be independently traced.

**Compliance Statement.**  This disclosure follows ICLR's LLM usage policy: the LLM acted solely as a *presentation aid*, not a source of scientific content. All theoretical claims are author-owned and can be validated from the text and the provided appendices.

