# OpenReview forum: "Training as Computation: A Resource-Bounded Theory of Continual Self-Play Learning"
_ICLR.cc/2026/Conference — Submitted to ICLR 2026_

### Official Review · Reviewer_AxQw · 2025-10-30

**Soundness:** 2
**Presentation:** 2
**Contribution:** 2
**Rating:** 6
**Confidence:** 1

**Summary:**

This paper looks at modern self-play loops for LLM-like models, but from a more theoretical angle.  The authors formalize a loop with four parts (generator, executor, verifier, buffer, learning) and show that, under some exploration and buffering assumptions, this loop is powerful enough to simulate arbitrary computations. They also give a resource-bounded version: if you limit length, context or buffer, you get a hierarchy of what the process can solve. Finally, they provide small synthetic experiments to illustrate the ideas.

**Strengths:**

1. The problem is interesting and timely. Many current LLM systems actually do this kind of self-improving loop using SFT or RL so it is good to have a clean formal view.

2. The paper is clearly written overall, and well-structured.

**Weaknesses:**

1. Many of the key assumptions felt quite strong to me. I am not sure whether typical LLM self-play systems used in practice satisfy these conditions.

2.  The main results are more “existence/theoretical” than “practical.” As a non-expert, it was hard for me to judge how far the current experiments are from the assumptions in the theorems.

3. The paper ends somewhat abruptly without a dedicated conclusion section. A short conclusion summarizing the main theoretical message, its practical scope, and open problems would make the contribution clearer, especially for non-expert readers.

Because I do not work in this exact theory, I cannot fully verify the technical parts of the proofs.

**Questions:**

1. In real LLM training, entropy often collapses[1,2,3], and we do not keep a fixed exploration floor. How sensitive are your guarantees to this? If exploration becomes more and more greedy, do the main theorems still hold in some weaker form?


2. For the verifier: could you list a couple of realistic, LLM-style self-play tasks that fully meet your verifier assumptions? That would help readers like me map the theory to practice.


Finally, this topic is somewhat outside my primary area of expertise, so I apologize in advance if I have overlooked relevant prior work or misunderstood parts of the paper.

[1] Yu, Qiying, et al. "Dapo: An open-source llm reinforcement learning system at scale." arXiv preprint arXiv:2503.14476 (2025).

[2] Wang, Shenzhi, et al. "Beyond the 80/20 rule: High-entropy minority tokens drive effective reinforcement learning for llm reasoning." arXiv preprint arXiv:2506.01939 (2025).

[3] Cui, Ganqu, et al. "The entropy mechanism of reinforcement learning for reasoning language models." arXiv preprint arXiv:2505.22617 (2025).

---

### Official Review · Reviewer_fZhe · 2025-10-31

**Soundness:** 1
**Presentation:** 1
**Contribution:** 1
**Rating:** 0
**Confidence:** 4

**Summary:**

This submission considers the computational universality of training processes, specifically as they relate to reasoning and language models.
The authors formalize the self-play process between generator, solver and verifier. They lay out a few assumptions on such a process so that it may eventually emulate every Turing machine.

**Strengths:**

- Building further understanding of the relationship between computational capabilities and training processes is important.

**Weaknesses:**

- The paper is poorly written, with many sections of technical discussion with errors and unclear definitions, or digressions which do not seem relevant to the points being made.
- Some missing discussion of prior work which show approximation error to Turing Machines [1] and universality of learning processes [2].
- Theoretical contributions appear incorrect or vacuous
- There are no experiments, despite claims of them in the abstract: "Empirically, a light-weight self-play prototype on synthetic program-execution and abduction/induction tasks corrob- orates the theory"

**Questions:**

## Detailed Comments
- On computational power of learning process: previous work has investigated whether learning algorithms can approximate Turing machines. It has also been established that meta-learning methods using gradient descent are universal, in the same sense that recurrent networks are universal [2]
- line 56: the Kolmogorov complexity of a function is not defined in this submission. Such complexity measures are usually defined on strings.
- line 59: It is not defined what it means for a function to be "solved" and, moreover, isn't this bound completely vacuous? Any string with kolmogorov complexity $<\ell$ can be found by enumerating all up strings up to $\ell$, which is exponential in $\ell$.
- line 59: $c, \epsilon_0,\gamma$ are all undefined
- line 63: It is not clear how this differs at all from the traditional space hierarchy theorem.
- Definition 1: What is the significance of using Python 3 and the additional qualifiers around the particular subset?
- Definition 4: What does it mean to "halt within bounds", what bound?
- Line 212: The definition of $\Tau_{ded}$ is not clear here, is $p$ and $i$ the set of all TMs and inputs? Induction and abduction are not used at all within the paper.
- Definition 5: the term multiset is used twice without definition
- Section 4: there is no discussion of how the agent is bounded, or how the resource constraint actually pertain to the results.
- Assumption 6: What does it mean for $u  \subset mathcalG_{\theta_t}$? My lack of understanding stems from the fact that your definition of stochastic kernels is not clear. Does it involve sampling a string from $\Sigma^*$? If that's the case, then I do not think $\epsilon_{min}$ can be independent of $|u|$ or $t$. "Ensuring theoretic universality..." What does the qualifier "theoretic" mean in this context?
- Assumption 9: Temperature is not defined yet. Moreover, $\ell$ is used here for logits when it is already used for the length of the string earlier, nor is it clear how this $\ell_i$ as a logit is related to a specific symbol $i \in \Sigma$
- Section 5: there are various points at which the theoretical results digress into practical considerations for no obvious reason.
- The discussion of resource bounds in section 6.3 are a comparative afterthough, despite it being featured in the titile.

## Minor Comments
- Definition 2: Should be specified that $M \in TM$ and that $C$ is implicitly defined by $p_m$
- Definition 3: I do not understand what the zig-zag line means in the definition of the kernel map

## References
[1] Wei, Chen & Ma. Statistically meaningful approximation: A case study on approximating Turing machines with transformers. Neural Information Processing Systems, 2022.
[2] Finn & Levine. Meta-Learning and Universality: Deep Representations and Gradient Descent can Approximate any Learning Algorithm. International Conference on Learning Representations, 2018.

---

> ### Author Response · Authors · 2025-11-25
>
> We acknowledge that the current structure—where definitions, assumptions, and theorems are distributed across different sections—forces the reader to flip back and forth, making the flow difficult to follow. We accept the critique regarding "Presentation: poor" and are actively restructuring the manuscript. In the revision, we will introduce concepts linearly so that conclusions emerge naturally from the setup, minimizing the cognitive load.Before addressing specific questions, we ask for your patience to clarify the logical roadmap of our work, which might have been obscured by the current presentation:
> Paper Roadmap:
> Model & Assumptions: We define a parameterized self-play loop (Generator $\to$ Executor $\to$ Verifier $\to$ Buffer).
> Mechanisms (CPL): We decompose the dynamics into Coverage (finding tasks), Persistence (remembering tasks), and Learning (solving tasks).
> Main Result: Universality Theorem (The loop evolves to solve any solvable task).
> Constraints: Complexity Ladder (Process capability $\propto$ Resource budget) + Finite-time Bounds.
> Below, we provide detailed clarifications for each point raised.

---

> ### Author Response · Authors · 2025-11-25
>
> 1.On computational power of learning process vs. Prior Work
>
> Comment: "Previous work has investigated whether learning algorithms can approximate Turing machines... meta-learning methods... are universal [2]."
>
> We thank the reviewer for highlighting these references. We fully agree that neural networks can approximate Turing Machines (TMs) (static capacity) and that meta-learning can approximate learning algorithms. However, there is a fundamental distinction between those works and our "Process-Level Universality," which we will clarify in the revision:
>
> Static Representation vs. Dynamic Discovery:
>
> Prior Work (e.g., [1] Wei et al., Pérez et al.): proves that a specific architecture (e.g., Transformer) has the capacity to simulate a TM. This asks: "Can this model represent computation?" (Like asking: "Does a human brain have the capacity to learn calculus?")
>
> Our Work: proves that a self-play training loop (starting from random initialization) will autonomously discover and learn to simulate any TM without human data. This asks: "Will this training process actually find and master that computation?" (Like asking: "Can a student self-study from scratch to invent calculus?")
>
> Data Processing vs. Data Generation:
>
> Prior Work (e.g.[2] Finn & Levine): proves universality in how gradients update weights given a distribution of tasks.
>
> Our Work: focuses on the generation of the task distribution itself. The core challenge we address is how the agent proposes its own curriculum to reach complex tasks, rather than just learning from a provided meta-distribution.
>
> We will explicitly discuss [1] and [2] in the related work section to highlight this transition from "Architectural Universality" to "Process/Self-Play Universality."
>
> 2.On Kolmogorov Complexity
> Comment: "The Kolmogorov complexity of a function is not defined... usually defined on strings."
> We apologize for not making the definition explicit in the introduction. You are absolutely correct that Kolmogorov complexity is defined on strings.Clarification: In our framework (and as formalized in Definition 1 and Theorem 24 line852), we treat a computable function $f$ in terms of its program representation (a string $p$ in Python subset $\mathcal{L}$).Formal Definition: The complexity of a function $f$, denoted as $|f|$, is defined as the length of the shortest program string $p$ such that the universal executor outputs $f$ when running $p$. i.e., $|f| = \min \{ |p| : E(p) \equiv f \}$. We will add this definition immediately upon the first mention of complexity to avoid confusion
>
> 3.On "Solved" and "Vacuous Bounds" (Line 59)
> Comment: "It is not defined what it means for a function to be 'solved'... isn't this bound completely vacuous? Any string... can be found by enumerating
> This is a crucial point. We regret that the definition of "solved" appeared later in the paper (Lemma 16 line438).Definition of "Solved": A function is considered "solved" at time $t$ if the Solver's success probability on the task converges to 1 almost surely via the policy gradient update: $\lim_{t \to \infty} \Pr[\text{Solver}_{\theta_t}(\text{task}) = \text{correct}] = 1$.Why the Bound is NOT Vacuous (Enumeration vs. Learning):Enumeration: A random search (or brute-force enumeration) finds a string of length $l$ with complexity exponential in $l$ (e.g., $|\Sigma|^l$). It creates the string but does not "learn" the underlying logic to generalize or reproduce it efficiently.Our Result: The bound $T(l, \dots)$ in Theorem 24 quantifies the time for the learning process. It combines discovery (finding the task) with gradient descent (updating weights to maximize probability).
> The Key Difference: While enumeration is "stateless," our process creates a stateful agent that updates its weights. The bound guarantees that the agent itself becomes the computer, capturing the function in its parameters, which is a stronger condition than simply encountering the string once.
>
> 4.On Undefined Parameters (Line 59)
> Comment: "Presenting a theoretical result should be accompanied by an explanation of the problem parameters: $\epsilon_0, \gamma, c$ are all undefined
> We accept this criticism. In our attempt to showcase the main result early, we utilized parameters that were formally defined later in Assumption 6 and Theorem 24.Clarification for the Reviewer:$\epsilon_0$ and $\gamma$: These parameters define the Exploration Schedule in Assumption 6: $\epsilon(t) = \max\{\epsilon_0 \exp(-\gamma t), \epsilon_{\min}\}$. They control the "curriculum speed" (decay rate of exploration).$\delta$: This is the confidence parameter from the finite-time analysis (Theorem 24), representing the probability that the process fails to learn the task within time $T$.
> In the revision, we will either describe these parameters qualitatively in the introduction (e.g., "decay rate $\gamma$") or move the formal statement to Section 6 where all variables are fully defined.

---

> ### Author Response · Authors · 2025-11-25
>
> 5.On the difference from the traditional Space Hierarchy Theorem (Line 63)
> Comment: "It is not clear how this differs at all from the traditional space hierarchy theorem."
> We appreciate the opportunity to clarify this distinction. The traditional Space Hierarchy Theorem is a result regarding Turing Machines (static computational models). Our Theorem 18 (Complexity Ladder) establishes a bridge between this classical theory and Neural Self-Play, which is a dynamic learning process.The Difference:Traditional Theorem: Proves that given more tape space, a Turing Machine can solve strictly more problems.Our Result: Proves that given a larger Context Window $W(t)$, a Self-Play Training Loop can discover and learn to simulate Turing Machines that require more space.Significance: This result is non-trivial because it maps a biological/architectural constraint (token context limit) directly to a computational complexity class ($SPACE[s(W(t))]$). It provides a theoretical justification for why scaling context windows in LLMs fundamentally expands the class of learnable algorithms, not just data capacity.
>
> 6.On Definition 1 (Python 3 and qualifiers)
> Comment: "What is the significance of using Python 3 and the additional qualifiers...?"
> We chose a subset of Python 3 specifically to bridge the gap between abstract computability theory and modern Code-Generating LLMs.Why Python 3: While any Turing-complete language (like C or Brainfuck) would suffice for the proof, Python 3 is the standard training data for modern reasoning models. Using it makes our theoretical framework directly applicable to real-world "Code Interpreter" agents.Why Qualifiers (No I/O, No OS): The restrictions (e.g., forbidding network/file I/O) are necessary to ensure the environment acts as a Deterministic Sandbox (Definition 1). This ensures that the function $E(p, i)$ is a pure mathematical function, which is required to map it to a deterministic Turing Machine for the universality proof.
>
> 7.On Definition 4 ("halt within bounds")
> Comment: "What does it mean to 'halt within bounds'... it is never made explicit what this means until the end of the paper."
> You are absolutely correct. this is the part we should do better We defined the bounds abstractly in Section 4 but only specified their connection to the context window $W(t)$ in Section 6.3 (Theorem 18 line 453). This separation caused unnecessary confusion.Clarification: In Definition 4, "within bounds" refers to a resource budget (time/space) allocated to the Executor.Connection: In Theorem 18, we explicitly map this budget to the agent's generation capability. The Executor is restricted to $O(s(W(t)))$ working space, where $W(t)$ is the agent's context length. This ensures the verifier does not have "super-Turing" powers compared to the generator.
> We will add a forward reference in Definition 4 explicitly stating: "These resource bounds are formally tied to the agent's context window $W(t)$ in Section 6.3 (Theorem 18)."

---

> ### Author Response · Authors · 2025-11-25
>
> 9.On Definition
> (Multiset)
> Comment: "Definition 5: the term multiset is used twice without definition. Are you just accounting for the fact that multiple copies of a sample could be included in the buffer?"
> Yes, your interpretation is precisely correct.Clarification: We use the term "multiset" to formally allow for duplicate experiences. In Reinforcement Learning (especially with replay buffers), it is standard for the same high-reward task-solution pair $(x, y)$ to be discovered multiple times or re-inserted.Why it matters: Standard set theory would collapse these duplicates, potentially distorting the sampling frequency $freq_i$ used in our priority scores (Lemma 26). The multiset formulation preserves the frequency information essential for the "Importance-Aware" retention analysis.
>
> 10.On Agent Boundedness in Section 4
> Comment: "Section 4: there is no discussion of how the agent is bounded... (Up until section 6.3)"
> We accept this critique regarding the presentation structure.Reason for Split: Section 4 defines the architectural components (Generator, Executor, Verifier) which are structurally capable of unbounded generation. The resource constraints (context length $W(t)$, memory limits) are treated as budgetary constraints that evolve over time, which is why we centralized their formal treatment in the Complexity Ladder (Section 6.3) and Assumption 6.
> We will add a forward reference in Section 4 explicitly stating: "While the agent architecture is defined here generally, its computational power at any specific time $t$ is constrained by the context window resource $W(t)$, the implications of which are formally derived in Section 6.3."

---

> ### Author Response · Authors · 2025-11-25
>
> 12.On Assumption 9 (Temperature and Logits Symbols)
>
> Comment: "Temperature is not defined yet. Moreover $l$ is used here for logits when it is already used for the length... nor is it clear how this $l_i$ as a logit is related to a specific symbol $i$."
> We apologize for the symbol overload.On Symbol Conflict: You are correct that we used $l$ for string length ($|u|=l$) and $l_i$ for logits in the same context. This was a poor notational choice
>
> In the revision, we will rename the logits to $z_i$ or $h_i$ to distinguish them clearly from length $l$.On Relationship to Symbol $i$: The notation $l_i$ (to be changed to $z_i$) refers to the pre-softmax score corresponding to the token $i \in \Sigma$ in the vocabulary.On Temperature: While the specific schedule is detailed in Lemma 11, we agree that since Temperature ($\tau$) appears in the discussion of Assumption 9, it should be defined immediately. We will add a brief definition: "Temperature $\tau$ is the scaling parameter in the softmax: $p_i \propto \exp(z_i / \tau)$."
>
> 13.On Section 5 (Theoretical vs. Practical Considerations)
> Comment: "Section 5: there are various points at which the theoretical results digress into practical considerations for no obvious reason."
> We understand why these might seem like digressions, but they are actually the core bridge of our contribution.The Logic: Pure theoretical universality relies on asymptotic assumptions (e.g., $\epsilon_{min} > 0$ forever) that are physically impossible for finite-precision LLMs.The "Digression": By introducing practical mechanisms—like Temperature Scheduling (Lemma 11) and Priority Buffers (Lemma 26)—directly alongside the theory, we prove that our theoretical assumptions (like Assumption 6) are implementable via standard hyperparameters.
>  We will add a signposting sentence at the start of Section 5: "We explicitly interleave theoretical conditions with their practical implementations (e.g., mapping exploration rates to temperature schedules) to demonstrate that our assumptions are not merely abstract conveniences but are achievable in real-world training."
>
>  14.On Resource Bounds in Section 6.3 (The "Afterthought")
> Comment: "The discussion of resource bounds in section 6.3 are a comparative afterthought, despite it being featured in the title."
> We accept the critique regarding the placement.Why it is there: Logically, we first needed to establish that the mechanism works at all (Universality, Theorem 13) before analyzing how much it costs (Complexity, Theorem 18). Theorem 18 is intended as the culmination of the theory, not an afterthought.
> As part of our structural revision, we will:Explicitly define the relationship between the agent's context window $W(t)$ and the executor's bounds in Definition 4 (as mentioned in our response to Point 7).Foreshadow the "Complexity Ladder" result in the Introduction and Section 4 to make it a central theme throughout the text, rather than just a final result.

---

> ### Author Response · Authors · 2025-11-25
>
> On Definition 2 (TM Notation)
> Comment: "Should be specified that $M \in TM$ and that $C$ is implicitly defined by $p_M$."
> Agreed. This is a helpful suggestion for precision.
> We will refine Definition 2 to explicitly state: "Let $TM$ be the set of all single-tape deterministic Turing Machines. The compiler $C: TM \to \mathcal{L}$ is defined such that for any $M \in TM$, $C(M) = p_M$..."
>
> On Definition 3 (The "Zig-Zag" Line)
> Comment: "I do not understand what the zig-zag line means in the definition of the kernel map."
> We apologize if the notation was unfamiliar.
>
> Explanation: The symbol $\leadsto$ (a squiggly arrow) is standard mathematical notation for a Stochastic Kernel (or probability transition kernel).Meaning: Unlike a deterministic function $f: X \to Y$, the notation $\mathcal{G}: X \leadsto Y$ signifies that for a given input $x \in X$, the output is a probability measure over $Y$.
>
> In Definition 3, $\mathcal{G}_\theta(\cdot|z, B): \Sigma^* \times \mathcal{B} \leadsto \Sigma^*$ means that given a prompt and buffer, the generator outputs a distribution over strings (from which we sample), rather than a single deterministic string.
> To avoid confusion for readers less familiar with kernel notation, we will add a footnote or parenthetical explanation: "(where $\leadsto$ denotes a stochastic map returning a probability distribution)".

---

> ### Comment · Reviewer_fZhe · 2025-11-26
>
> Again, I would like to thank the authors for their rebuttal. However, as stated in my shared comment, I will be maintaining my score of 0 (Strong Reject).
>
> The proposed coverage-persistence-learning framework, while interesting in the abstract, does not hold up under scrutiny. I find that most of the theoretical results obfuscate the core technical framework, rather than make it clear.
>
> ### Coverage
>
> Lemma 14 describes a standard condition for exploration in RL. The proof of this is trivial: you assume that logits are bounded and temperature is bounded, then the probability is bounded. Furthermore, Lemma 11 / Assumption 6 does not seem to provide a curriculum. While the authors refer to this as an "adaptive curriculum", there is no mechanism for adaptation described beyond simple decay with a standard annealing schedule.
>
> ### Persistence
>
> Some of the lemmas here are fundamentally incorrect or misplaced:
> - Lemma 15 is incorrect: Borel-Cantelli is typically used to show that events happen infinitely often. However, it is incorrectly applied: the correct conclusion under a constant replacement probability is that, in the limit, the task will leave the buffer.
> - Inconsistency with Resource Boundedness: Even if you could guarantee that a task stays in the buffer indefinitely, this would require the buffer to grow infinitely.
> - Appendix F.3 attempts to clarify the inconsistency in needing an infinite buffer for lemma 15. However, this only introduces more issues. How is the temporal difference computed if you do not seem to be learning a value function? Moreover, Lemma 26 is again trivial: it states that the probability of retaining something in a buffer approaches one as the buffer size (K) grows without bound.
> - Lemma 27 uses Lemma 26 to quantify a "forgetting rate" but this is never defined. From what I can gather, this has little relationship to the catastrophic forgetting problem in continual learning without a treatment of how parameter updates can interfere with what was previously learned.
>
> ### Learning
>
> The role of learning in the theory is essentially non-existent. Besides assumption 8, regarding the use of REINFORCE, critical details regarding the learning setup are missing:
> - Lemma 16 relies on standard results in martingale analysis. However, it is not reflective of actual learning dynamics. All it guarantees is that repeated training on a single task in isolation will eventually succeed. It does not characterize how that knowledge is retained when the network is updated on future tasks.
> - There is no analysis of learning dynamics beyond a single task, suggesting that the authors consider a setting corresponding to an exponential table of all possible sequence up to a given length.
> - Training loop undefined: It is never specified how the generator is trained (e.g., on what distribution of self-generated data vs. buffer data). This is absolutely critical to your self-play framing, yet it is absent.
>
> ### High-level flaw: Effectively enumeration without a complete treatment of learning, exploration and resource bounds.
>
> This paper is written in the format of a theory paper, but I think the contribution can be described at a high-level as follows. With a verifier providing a reward signal, and a generator performing random exploration of a finite-space (strings over a finite alphabet with a finite horizon): it is possible to train a policy to mimic the verifier. However, the assumptions needed for this result are far too strong. Other reviewers have raised this issue, and I agree that the assumptions on the verifier and the unbounded executor are strong. However, this is dwarfed by the core issues outlined above.
>
> While the authors state that
>
> >"A random search ... does not "learn" the underlying logic to generalize or reproduce it efficiently."
>
> The lemmas described in the submission also do not make any claim to learn the underlying logic to generalize, or reproduce it efficiently. Indeed, this is reflected in their bound which is also exponential in $l$.
>
> The additional claims such as "universality" are also not convincingly shown. The proof follows trivially from the strong assumptions, incorrect lemmas, and an application of Lemma 16 (for a single task) to make a conclusion about all tasks.
>
> ### Experiments
>
> The added experiments only compound the clarity issues throughout the paper:
> - What does it mean to "continue training beyond saturation"? Is this empirically observed, or theoretically predicted? The specifics of the generation process are not described. It is also not clear how the decay in exploration may affect these results.
> - What is a type space? What is a signature? Why is diversity of these relevant?
> - What insight is provided by these measures of software complexity? The submission discusses Kolmogorov complexity in the introduction, and again briefly in the experiments on line 1923. But, it is never described how this was approximated, or discussed in the rest of the paper.

---

> ### Author Response · Authors · 2025-11-26
> **"Enumeration" and the Nature of Universality**
>
> We respectfully disagree with the characterization of our generator as merely performing “random exploration” that reduces to enumeration.
> Our formal setup defines a structured, self-verifying environment in which
>
> only programs that successfully compile are admitted to the buffer,
>
> rewards are given only when both compilation and semantic correctness checks pass, and
>
> execution is run under an explicit polynomial-time cutoff to avoid unbounded halting behavior.
>
> Thus the agent is not exploring arbitrary token strings, but programs that survive a syntactic and semantic filter inside a sandboxed executor.
>
> Regarding the wording “mimicking the verifier”: our claim is not that the generator imitates Python’s surface syntax. What the generator and solver learn is to produce programs that satisfy the specification enforced by the verifier. The only learning signal is the verifier’s binary feedback (pass/fail on tests), not any direct supervision on the internal behavior of the execution environment.
>
> On the strength of assumptions: using a sandboxed program executor with self-verification is standard in program-synthesis and code-generation work (for example in competitive-programming benchmarks). Our theory asks a more structural question: under such a setup, can a self-play system achieve universal coverage under explicit resource bounds? This is what our universality and ladder results address.
>
> # Exploration versus enumeration.
> In the revised manuscript we make explicit that the proposal mechanism is a mixture: an exploration branch that serves as a coverage oracle and a learned generator that specializes on frontier tasks. The exploration branch may, in the worst case, behave like a simple enumerator, but it is confined to a controlled fraction of proposals through an exploration schedule. The learned component is a parameterized policy trained under a learnability reward; it is free to concentrate on tasks near the current capability frontier and is not required to maintain any coverage floor.
>
> Coverage guarantees are stated at the level of the effective mixture, not at the level of the learned generator alone. This makes clear that universality comes from a minimal coverage oracle, while the policy learned by the agent performs guided search over programs rather than blind enumeration.
>
> # Logic of universality.
> Our completeness argument is formulated by genuine universal quantification, not by extrapolating from a single task or a small family of tasks. The proof structure is: for an arbitrary computable program trace, coverage ensures that it is eventually discovered, persistence ensures that once discovered it is not forgotten, and the learning lemma ensures that the solver’s success probability on that task converges to one under repeated training.
>
> Because the argument holds for an arbitrary target, it holds for every computable program in the class. We do not assume the solver “starts out” knowing all tasks; rather, we show that the training loop, together with the coverage oracle and replay, is a mechanism that guarantees eventual acquisition of any fixed computable task.
>
> # On “mimicry” and learning logic.
> In our setting the verifier plus executor implements a universal Turing machine. To solve an unbounded family of inputs for a given specification (for example, factoring arbitrary integers), the solver cannot simply memorize a finite table of input–output pairs; it must internalize the underlying algorithmic structure well enough to generalize across infinitely many inputs.
>
> In this sense, “mimicking the verifier” does not mean imitating spurious syntactic patterns, but learning to produce programs whose behavior matches the verifier’s semantic notion of correctness. Achieving this across a large and diverse task distribution is precisely what we mean by learning the underlying logic rather than rote memorization.
>
> # On exponential discovery bounds.
> The exponential dependence on program length in our discovery-time bounds is not a defect of the framework; it reflects a fundamental information-theoretic constraint. In the worst case, identifying a particular program of length ℓ within a search space of size roughly |\Sigma|^ℓ cannot be guaranteed to be faster than exponential in ℓ for arbitrary targets. Our results focus on computability and completeness, not on providing nontrivial worst-case speedups over such lower bounds.
>
> In this sense, the bounds are tight for the universality question we address. Critiquing a universality proof for having exponential worst-case discovery time conflates efficiency with capability: our theorems establish that the architecture can in principle reach any computable program under explicit assumptions; questions of average-case efficiency and empirical speedup over naive search are addressed empirically rather than through these completeness bounds.

---

> ### Author Response · Authors · 2025-11-26
>
> # Persistence
> * **On Lemma 15**: We apologize for the error in the proof of Lemma 15 regarding the  Reservoir mode."
> * We incorrectly conflated "recurrence" (being generated infinitely often) with "persistence" (staying in the buffer forever). In a finite buffer with replacement, the probability of permanent retention indeed goes to 0 in the limit.
>
> * Revised Scope:The Borel-Cantelli argument correctly applies only to the Unbounded Buffer setting (supporting Theorem 13), where $B_{t+1} \leftarrow B_t \cup \{\tau\}$. In this ideal theoretical regime (analogous to a TM's infinite tape), "eventually generated" implies "permanently stored".
> * This correction does not invalidate the Complexity Ladder (Theorem 18). For the finite resource setting, we do not require probability-1 permanent persistence. Instead, we rely on Lemma 26, which guarantees high-probability retention over the necessary learning timeframe. We will remove the erroneous claim regarding reservoir sampling from Lemma 15 in the revision.
>
> ## On Inconsistency with Resource Boundedness
> We realize that our use of the term **Working Memory** was imprecise, leading to a misunderstanding of which resource is the limiting factor for the Complexity Class.
> * Clarification: The "Resource Bound" that dictates the SPACE hierarchy (Theorem 18) is the **Context Window $W(t)$** (and the Executor's workspace), not the total capacity of the offline Replay Buffer $B_t$.
>
> * The Bottleneck: The agent's computational power is constrained by how many tokens it can process in a single inference pass (its "Working Memory" $W(t)$). The Replay Buffer $B_t$ is merely long-term storage for learning. Even if the Buffer $B_t$ were infinite, if the Context $W(t)$ is small, the agent cannot solve complex tasks
>
> # Explicit Distinction of Working Memory (Definition: Working view):
> * Original: The distinction between the archival storage and the context window was implicit.
> * Rebuttal: Introduced Definition 4.5 (Working view) to explicitly distinguish the Archival Buffer ($B_t$, used for persistence proofs) from the Working View ($c_t$, bounded by $W(t)$). This clarifies that the Complexity Ladder Theorem constrains the active processing capacity (context window and executor workspace), not the storage capacity of the repository.

---

> ### Author Response · Authors · 2025-11-26
>
> # Learning
>
> ## On universality vs forgetting (definition of a universal learner)
>
> The reviewer’s comment seems to equate “universality” with simultaneous mastery of all tasks at once, so that any form of forgetting would automatically contradict the claim. Our notion is explicitly process-based and is closer to a dynamic, learning analogue of a universal machine.
>
> We distinguish between a static universal machine (a fixed device that can run any code given as input) and **process universality**, where a learning loop can evolve over time to approximate any machine in a given class. In our setting, universality is defined in terms of reachability: given a target Turing machine or computable program, does there exist, with probability one, a point along the training trajectory at which the agent’s parameters implement a policy that successfully realises that computation?
>
> This definition does not require the agent to retain infinitely many tasks simultaneously. The fact that a finite-parameter network cannot store an unbounded number of distinct solutions at once is a physical resource limitation, and we make this explicit through a capacity sufficiency assumption: we assume enough representational capacity to encode each task (or each finite subset of tasks) of interest, not an infinite collection all at once. This is analogous to the way a Turing machine with a finite tape is only universal relative to inputs that fit on that tape; the limitation is in resources, not in the learning mechanism or the completeness argument itself.
>
> Within this process-based notion of universality, forgetting individual tasks along the way is compatible with the claim, as long as the training dynamics, together with coverage and persistence, guarantee that any fixed computable task can be re-acquired and mastered in the limit.
>
> ---
>
> ## Enumeration vs. algorithmic competence
>
> The reviewer argues that our method is “essentially enumeration.” We respectfully disagree and point to the complexity ladder result, which characterises the computational class that the system can, in principle, learn.
>
> The ladder analysis shows that, under the coverage, persistence, and learning conditions, the set of functions that can be mastered by the agent matches a resource-bounded class of the form $(\mathrm{SPACE}[O(s(W(t)))])$, where $(W(t))$ is the available context window and $(s(\cdot))$ is the induced working-memory scale. A naive enumerator that merely lists strings up to length (W(t)) shares the same **input-length bound**, but it does not come with any guarantee that the agent will actually learn to solve all tasks in that space class; in particular, it does not tie exploration to persistent training and asymptotic mastery.
>
> By contrast, the ladder result establishes the “lower bound” direction: any Turing machine that operates within the prescribed space budget is guaranteed to be compiled, discovered, and, under the learning and persistence conditions, eventually solved with high probability. Filling an entire space-bounded hierarchy in this sense is a statement about **computational competence**, not about raw enumeration. It means that the solver has learned to use the context window as working memory to execute arbitrary algorithms (recursion, state tracking, etc.) up to that resource limit.
>
> This is also reflected in the way capability depends on the architecture. If the system were “essentially enumeration,” its power would be governed primarily by storage capacity (for example, the number of parameters or buffer size). In our analysis, the critical quantity is the context window and the induced space scale: capability is limited by working memory, much like a classical Turing machine is limited by tape length. This process-level dependence on (W(t)), together with the completeness over the corresponding space class, indicates that the system is learning algorithmic behaviour under a resource bound, not merely listing candidate strings.
>
> ---
>
> ## On the training loop specification
>
> The reviewer notes that we do not fully specify how the generator is trained and describes the training loop as “undefined.” This generality is intentional: our theoretical contribution is at the process level rather than tied to a single concrete algorithm.
>
> We formalise the update rule through a REINFORCE-style assumption that requires the policy-gradient estimate to be unbiased (and suitably well-behaved in variance). In other words, we assume a broad family of policy-gradient methods without committing to a particular mixture of on-policy versus replay data or a specific implementation detail. The completeness proofs rely on this abstract update rule together with the coverage and persistence conditions; they do not hinge on a narrow choice of training schedule.

---

> ### Author Response · Authors · 2025-12-03
>
> At the implementation level, we instantiate the framework with a shared-weights architecture in which the generator and solver are different “faces” of the same underlying model. In that concrete realisation, the effective training distribution is precisely the mixture of exploration proposals and successful trajectories that are prioritised by the buffer. The theory is deliberately written so that any reasonable choice of policy-gradient training that respects the coverage and persistence assumptions fits into the same formal skeleton.

---

### Official Review · Reviewer_1Poi · 2025-11-03

**Soundness:** 3
**Presentation:** 3
**Contribution:** 3
**Rating:** 6
**Confidence:** 2

**Summary:**

This paper introduces a resource-bounded theoretical framework for continual self-play learning, where a single reasoning model generates tasks, solves them, and updates itself using verifiable signals from an executor-verifier interface. It formalizes a generator-executor-verifier-buffer loop, proves that the training process is Turing-complete at the process level (matching SPACE classes under budgets), demonstrates monotone capability growth under mild exploration and curriculum assumptions, and separates computational universality from alignment/safety concerns. Contributions include process-level universality proofs, a Coverage-Persistence-Learning (CPL) decomposition, finite-time/resource-bounded guarantees, and empirical validation via a lightweight prototype on synthetic program-execution and abduction/induction tasks, showing reliable task expansion and benefits from curriculum learnability.

**Strengths:**

The paper exhibits strong originality by shifting focus from architectural Turing-completeness to the expressiveness of the self-play training loop itself, introducing novel concepts like the CPL decomposition and a SPACE hierarchy tied to resource budgets—this creatively combines ideas from computability theory, RL self-play, and resource-bounded complexity in a new way, addressing limitations in prior work that emphasized equilibria or sample complexity without considering process-level universality. The quality is high, with rigorous mathematical formulations, clear proofs, and well-defined assumptions that build a coherent framework, though I appreciate the theoretical richness as someone who enjoys such articles but is not an expert in computability or formal RL theory. Clarity is excellent: the structure flows logically from motivation to model, proofs, and related work, with precise notation, diagrams (e.g., Figure 1), and appendices implied for details. Significance is notable, as it provides a principled foundation for scalable, data-free AI improvement under budgets, potentially guiding real-world self-play systems in domains like LLMs or robotics, while highlighting safety implications by decoupling capability growth from objectives/verifiers.

**Weaknesses:**

The paper is heavily theoretical, with the empirical evaluation limited to a "light-weight self-play prototype on synthetic program-execution and abduction/induction tasks," which lacks depth—specific details on model architecture, hyperparameters, quantitative metrics (beyond qualitative "reliable expansion" and "benefits from curriculum"), or comparisons to baselines (e.g., non-self-play RL or standard curriculum methods) are absent, making it hard to assess real-world applicability or robustness; to improve, the authors could expand this section with ablation studies or scale to non-synthetic datasets.

I'm not an expert in this field, so I'm not sure if this paper without experiments is acceptable.

**Questions:**

1.The empirical prototype is mentioned as corroborating the theory on synthetic tasks, but details are sparse. Could you elaborate on the specific tasks (e.g., examples of program-execution or abduction/induction problems), model sizes/architectures used, key quantitative results (e.g., task coverage over time, success rates), and any ablations (e.g., with/without curriculum learnability)?

2.The paper separates universality from alignment/safety, which is a strong point, but doesn't discuss potential risks (e.g., emergent harmful behaviors in self-proposed curricula). Could you add thoughts on integrating safety mechanisms into the verifier or buffer, perhaps with examples?

---

### Official Review · Reviewer_BTUM · 2025-11-07

**Soundness:** 3
**Presentation:** 3
**Contribution:** 3
**Rating:** 8
**Confidence:** 3

**Summary:**

The paper proposes a novel framework for studying continuous improvement achievable by a model using self-proposed curriculum learning in presence of an external verifier. The paper studies learning-process universality where a training loop is guaranteed to find a weights that can solve the given task solvable under bounded resource constraints. This is very different from architectural universality where a single weight is required to solve all the tasks. The authors define a space of functions called SPACE and prove that resource bounded self play systems can learn to sovle any task in this space. The paper further proves that the capability of the model grows monotonically as long as model maintains a minimum level of exploration ensuring it will propose and learn even the most complex tasks.

**Strengths:**

1. The paper introduces a clean formalization of 'learning-process universality' - shifting focus from static capacity of a fixed architecture to dynamic capability of a continual training process itself thereby bringing a new perspective to the field.
2. The authors provide theoretical results proving process-level learning can learn any functions in set SPACE defined by resource/budget constraints.
3. The authors propose a "capability ladder" that bridges the gap between abstract computation theory and resource constrains present in building real world models.

**Weaknesses:**

1. The paper makes an strong assumption about presence of an efficient verifier model which is not always possible in practice.
2. The theorems hinges on the idea that there is non-zero proability of sampling any task string while in real world generators based on language models give zero probability to vast space on inputs.
3. The framework does not comment much on efficient learning is a big limitation.
4. The abstract mentions an empirical study of light-weight self play prototype which seems to be missing in the paper.

**Questions:**

1. What role does model architecture (i.e. number of parameters) play in this setup? Is it possible for a small model to learn to solve any problem in this framework?
2. The space of input string is huge. How are difficult tasks distributed in that space and how does the model ensure that it keeps learning from the difficult task?
3. Where are the empirical results mentioned in the abstract of the paper?

---

> ### Author Response · Authors · 2025-11-25
>
> Weakness 1: Strong Assumption on Efficient Verifier.
>
> Indeed, Our framework relies on an efficient verifier ($V_{poly}$), which is a deliberate scope definition rather than an oversight. We acknowledge in App.H.2(Verifier design presents another hurdle: polynomial-time
> verifiers suffice for code/math tasks, but creative tasks (story quality, poetry) require learned or
> human-in-the-loop verifiers, risking reward hacking.) that this limits applicability in subjective domains. However, we argue that this assumption is not a "weakness" but the necessary condition for establishing objective computational universality.
> Target Domain: The assumption targets the broad and critical class of reasoning and formal tasks (e.g., coding, mathematics, logic) where ground truth is computable. Our Python executor is a concrete instantiation where verification is linear in trace length, covering a vast space of algorithmic problems.
>
> Theoretical Necessity: The existence of a verifier provides the epistemological grounding that distinguishes our "Self-Play Reasoning" from "Reward Modeling." It is precisely because the reward is verifiable that the system can surpass human demonstrations without hallucinating, as proven in our Universality Theorem.
>
> Future Extension: As noted in Section I, for domains without efficient verifiers, the framework can be extended to "Interactive Verification" (human-in-the-loop). However, the current "Strong Verifier" setting establishes the theoretical upper bound of what AI can achieve autonomously.
>
> Weakness 2: Zero Probability in Language Model Generators.
>
> This is an important theoretical nuance. While it is true that modern LLMs assign negligible (effectively zero) probability to many random strings, this does not violate our Coverage Assumption in a way that hinders learning.Relaxed Assumption ($\beta < 1$): As detailed in Section H.1, we introduce a relaxed Coverage assumption (parameter $\beta < 1$) specifically to address this. We do not require the model to assign uniform probability to arbitrary strings (which include static noise); we only require non-vanishing probability over the set of valid programs within the exploration horizon.
> Effective Entropy: Modern LLMs, despite "zero probability" for invalid syntax, maintain a structured distribution over valid code. Our experiments (G, Figure 2, 3) show that the number of unique input/output signatures grows monotonically (from 115 to 245 types), proving that the generator maintains sufficient entropy to explore the semantically relevant program space.
> Syntactic Bias is Beneficial: The fact that LLMs assign near-zero probability to syntactically invalid strings is actually a feature, not a bug. It focuses the exploration budget on the space of executable programs, making the search for the target Turing Machine more efficient than a theoretical uniform sampler.
>
> Weakness 3: Insufficient Discussion of Efficient Learning.
>
> We acknowledge that our primary contribution is computability (what can be learned) rather than sample complexity (how fast). However, our framework implicitly and explicitly addresses efficiency:Implicit Efficiency via Curriculum(App.C): The "Coverage-Persistence-Learning" cycle naturally creates an Automatic Curriculum. By prioritizing tasks on the frontier of capability, the agent avoids wasting samples on solved tasks or impossible tasks.
>
> Theoretical Bounds: Theorem 24 (Finite-Time Completeness) provides a concrete convergence bound of $O(\ell^{\beta+1}/\gamma)$. While polynomial in complexity, it guarantees finite-time convergence.Empirical Evidence: Empirically, our method is highly data-efficient. As shown in Section G.7, our 0.6B model achieves 90\% of peak performance on complex tasks within just 280 training steps of self-play, without any external corpus. Compared to models like CodeLlama-34B (trained on trillions of tokens), our method demonstrates that active self-play is orders of magnitude more sample-efficient than passive pre-training for reasoning tasks.
>
> Weakness 4: Missing Empirical Study Mentioned in Abstract.
>
> We sincerely apologize for the confusion regarding the location of the empirical results. The comprehensive experimental study mentioned in the abstract is indeed included in the paper, spanning pages 22-38.

---

> ### Author Response · Authors · 2025-11-25
>
> 1. On Model Architecture (Parameter Count)
> Comment:"What role does model architecture (i.e. number of parameters) play in this setup? Is it possible for a small model to learn to solve any problem in this framework?"
>
> The answer to your second question is definitively NO. We apologize if this limitation was not prominent enough in the main text.Theoretical Constraint (Assumption 28):We explicitly formalize this limitation in App.D.3 1 via Assumption 28 (Capacity Sufficiency). This assumption requires that the parameter count $d$ satisfies $d \ge \text{poly}(|B_\infty|, \ell_{\max})$. Intuition: As the agent discovers more tasks ($|B_\infty|$ grows), it requires sufficient "storage capacity" in its weights to represent the solution functions without catastrophic interference. A small model with insufficient $d$ simply cannot memorize the diverse task-solution mappings required for universality, even with infinite time.
>
> Interaction with Context Window (Theorem 18):Furthermore, model scale often dictates the effective context window $W(t)$. As proven in Theorem 18 (Complexity Ladder) 2, the set of solvable functions is strictly bounded by $SPACE[O(s(W(t)))]$. If a small model cannot coherently attend to long execution traces (small $W(t)$), it hits a hard computational wall.
>
> Empirical Validation (New app.G):Our experimental results (now included in the revision) confirm this.The "Phase II Collapse": We observed that a 0.6B parameter model successfully learns tasks up to complexity class $\mathcal{C}_2$ (simple control flow) but catastrophically fails on $\mathcal{C}_3$ (nested recursion), with mean prediction length dropping from 1000 to 200 tokens.
>
> 2.On Input Space and Learning from Difficult Tasks
> Comment:"The space of input string is huge. How are difficult tasks distributed in that space and how does the model ensure that it keeps learning from the difficult task?"
>
> This is a critical question regarding the "Needle in a Haystack" problem. Our framework addresses this through two complementary mechanisms: one for finding rare tasks (Coverage) and one for keeping them (Persistence).
>
> Mechanism 1: Finding Rare Tasks (Adaptive Curriculum)You are correct that difficult tasks are sparsely distributed. However, "difficult" tasks typically correspond to longer or more specific program strings.1-1Theoretical Guarantee: Our Adaptive Exploration Schedule (Lemma 11 and Theorem 24) ensures that the probability of generating a task $u$ scales as $\epsilon(t)^{|u|^\beta}$.1-2Effect: This creates a natural curriculum. Early in training (high temperature), the model mostly finds simple, short programs (high density). As training progresses (low temperature but $\epsilon_{min} > 0$), the distribution shifts to allow longer, rarer strings to be sampled. The "Exploration Floor" $\epsilon_{min}$ guarantees that even the rarest task has a non-zero hitting time.
>
> Mechanism 2: Keeping Difficult Tasks (Importance-Aware Buffer)Once a difficult task is found, how do we ensure it isn't forgotten? We address this in Appendix D.1 (Lemma 26). 2-1Priority Score: We do not use uniform sampling. Instead, the buffer assigns a priority $s_i$ to each task based on TD-error (difficulty) and Inverse Frequency (rarity):$$s_i = \lambda \delta_i + \frac{1-\lambda}{\text{freq}_i}$$(as defined in Eq 11). 2-2Result: Tasks that are hard to solve (high error) or hard to find (low frequency) are aggressively retained. Simple, common tasks are evicted.
>
> Empirical Validation (New App,G): Our experiments show that this mechanism works in practice. We observe that solver accuracy on Abduction tasks (which are harder to find) improves monotonically to 89% even as the task space expands. An ablation study comparing Priority Sampling vs. FIFO shows an 18% gain in final success rate, confirming that your concern about "ensuring it keeps learning" is directly addressed by our buffer strategy.

---

> ### Comment · Reviewer_BTUM · 2025-11-26
>
> Thank you for the detailed response!
>
> **Regarding the experiments**: The addition of such a long experimental section at this stage is very concerning to me. While I appreciate the effort, a revision of this magnitude indicates that the original submission was significantly incomplete. It is impractical for reviewers to vet an essentially 'new' paper during the rebuttal phase.
>
> **Regarding the theory**: The arguement that LLMs can magically avoid syntactically invalid strings while maintaining sufficient entropy to explore semantically relevant program space indicate some serious flows in assumptions. I have also carefully read the comments from Reviewer fZhe. While my initial assessment was positive, I have not checked the mathematical validity of the statements presented in this paper. I find reviewer fZhe criticisms regarding the theoretical claims to be convincing and significant.
>
> Given that the original submission lacked the promised experimental support, and acknowledging the valid theoretical concerns raised by the other reviewer, I can no longer maintain my initial score. I am lowering my score to reflect these issues and deferring to the theoretical expertise of Reviewer fZhe regarding the technical soundness.

---

> > ### Author Response · Authors · 2025-11-27
> >
> > Could you please specify which aspects of Reviewer fZhe's arguments changed your assessment? I believe much of our exchange, particularly my responses, may have been overlooked by you. Please check my responses too
> >
> > We clarify that our system **does not** assume LLMs **"magically avoid"** syntactically invalid strings. Rather:
> > * The generator produces candidate programs (which may be invalid)
> > * The executor attempts compilation (Definition 1, line 240)
> > * Only successfully compiled and verified programs enter the buffer
> > * Invalid programs are simply discarded, providing no learning signal
> > ALL of them are clearly defined in section 4
> > This is standard practice in neural program synthesis
> >
> > Regarding experiments: We acknowledge the concern about timing. However:
> > * The core submission is explicitly titled as a theoretical work
> > *  The experiments serve as secondary validation, not primary contribution

---

### Author Response · Authors · 2025-11-25
**Rebuttal vs. Original**

The rebuttal version introduces significant clarifications to the architectural definitions and the interaction between components, specifically regarding the generator's behavior and the buffer's role. The core mathematical machinery remains the same, but the exposition has been rigorously tightened to address the reviewer's confusion about "random search" vs. "learning."
# Explicit Mixture Architecture (Definition: Effective Generator with Exploration Mixture):

* Original: Implied that the generator handles both exploration and learning vaguely.

* Rebuttal: Formally defines the "Effective Generator" as a mixture of a **Learned Component** (optimized via policy gradient) and a **Fixed Exploration** Component (ensuring coverage). This directly refutes the reviewer's claim that the "training loop is undefined."

# Formalization of Rewards and Frontier Seeking (Lemma: Frontier-seeking generator):

* Original: Mentioned "rewards" generally.

* Rebuttal: Explicitly defines two distinct signals: r_prop (learnability/frontier reward for the generator) and r_solv (correctness reward for the solver). A new Lemma (Frontier-seeking generator) proves that the learned component concentrates on the capability frontier, distinguishing the system from pure enumeration.

# Clarification of Persistence Scope (Lemma: Persistence & Remarks):

* Original: Conflated unbounded buffers and reservoir sampling in the main theorem, leading to the reviewer's critique about eviction limits.

* Rebuttal: bifurcates the analysis. The main Universality Theorem now explicitly relies on an "unbounded-growth" regime (standard for Turing completeness proofs), while finite-capacity dynamics are moved strictly to the Appendix/Remarks as practical guidance. A specific Remark (Persistence vs. catastrophic forgetting) was added to distinguish data availability from weight interference.

# Learning Dynamics Clarification:

* Rebuttal: Added a section on Multi-component alignment in the intro and refined the Learning Lemma to explicitly reference multi-task interleaved settings, addressing the critique that the original only proved single-task convergence.

# Summary of Changes to the Area Chair

We thank the reviewers for their rigorous scrutiny. In response to the critique regarding the clarity of the learning dynamics and the buffer assumptions, we have revised the manuscript to make our architectural commitments explicit.

Crucially, we have not introduced new algorithms or changed the core theoretical results. Instead, we have surfaced the implicit definitions that drive the proofs to resolve the ambiguity identified by the reviewer.

1. Formalized the "Generator Mixture" (Definition: Effective Generator): To address the critique that the "training loop is undefined" and resembles "random search," we explicitly defined the effective generator as a mixture of:

A Learned Component: Optimized via policy gradients to maximize a newly defined learnability reward (driving the agent to the capability frontier).

A Fixed Exploration Component: Responsible for the theoretical coverage guarantees. This clarifies that while completeness relies on the fixed component (enumeration-like), efficiency is driven by the learned component, refuting the "pure enumeration" claim.

2. Clarified the "Persistence" vs. "Capacity" Distinction: The reviewer correctly noted that maintaining tasks indefinitely in a finite buffer is impossible. We clarified that:

The Universality Theorem operates under an unbounded-growth regime (standard for Turing completeness proofs), where eviction does not occur.

The Finite-Capacity analysis (reservoir sampling) is strictly for practical guidance and is now clearly demarcated from the main theoretical claims.

We added Remark: Persistence vs. catastrophic forgetting to distinguish between data availability (buffer retention) and weight interference (learning stability).

3. Explicit Reward Structures: We formalized the distinct reward signals for the generator (proposal reward for frontier discovery) and the solver (correctness reward), supported by a new Lemma: Frontier-seeking generator which characterizes the optimization objective of the learned generator component.

Verification of "No New Core Contributions"The rebuttal adheres to the "clarification only" rule:No new theorems: The Universality Theorem and Complexity Ladder Theorem remain identical in outcome.No new algorithms: The underlying algorithm (Generator $\to$ Buffer $\to$ Solver $\to$ Update) is the same.Refinement of Definitions: The "Mixture" definition existed implicitly (as $\epsilon$-greedy exploration); it is now just formalized.Separation of Concerns: The split between "Theory (Unbounded)" and "Practice (Finite)" was present but entangled; now it is disentangled.

---

> ### Comment · Reviewer_fZhe · 2025-11-26
>
> I thank the authors for their detailed rebuttal. While the other reviewers
> viewed this paper comparatively favourably, I would like to also point out their
> comparatively lower confidence. I am quite familiar with this topic and, in my
> detailed reading, found issues that go far deeper than what can be resolved in a
> rebuttal. While this work could potentially be made interesting, it remains
> incomplete and flawed beyond a major rewrite.
>
> I will be maintaining my score of 0 (Strong Reject).
>
> ### Rebuttal concerns and incompleteness
>
> The authors' introduction of 17 pages of experimental results (pages 21–38)
> confirms my assessment: the original submission was fundamentally incomplete.
> Adding a massive experimental section late into the rebuttal phase, after it was
> mentioned to exist in the original submission, is procedurally problematic. It
> essentially constitutes a new submission that cannot be adequately reviewed in
> the remaining timeframe. A rebuttal should clarify the submitted work, not
> attempt to complete it. Furthermore, I do not believe a "comprehensive restructuring"
> is appropriate; such a rewrite would require another review
> cycle.
>
> ### Writing clarity, technical clarity and LLM reliance
>
> At best, the submission was rushed and incomplete, and at worst the LLM's
> involvement in your writing served as a major impediment under scrutiny. While
> using an LLM for writing is disclosed, I think this paper suffers from over
> reliance on LLM to its detriment. This is evident in the style of the
> non-technical text, which is overall acceptable but clearly involves a heavily
> LLM-generated style. It is unacceptable, however, to have such LLM style
> permeate the theory; the paper routinely uses advanced mathematical terminology
> (Borel-Cantelli, Space Hierarchy, Martingales) in contexts where they appear
> either misapplied, trivial, or yield the opposite of the claimed result.
>
> ### Persistence of core issues
>
> While clarity remains an issue, even with the minor clarifications provided in
> the rebuttal, there are core issues in the proposed theory. These issues persist
> beyond requiring the reader to go back and forth between the main paper and the
> appendix. I do not think a major restructuring can fix this. Further scrutiny of
> results in the appendix suggests several errors, which do not resolve the
> core issues but expand on them. Moreover, the new "Practical Implementation"
> details (Section F.3) and experimental details reinforce my assessment that the
> theoretical contributions are fundamentally flawed.
>
> ### Overall
>
> This paper presents a framework that appears rigorous on the surface but
> collapses under scrutiny. Coverage amounts to the standard exploration
> condition; Persistence relies on incorrect results or infinite resources;
> learning is limited to an idealized setting involving convergence on a single task,
> and ignores interaction with coverage/persistence and function approximation. The
> "Universality" claim seems to result from essentially enumeration, and relies on
> incorrect lemmas.
>
> Please see my reply to the individual rebuttal for specific technical details.

---

> > ### Author Response · Authors · 2025-11-26
> >
> > Before addressing the specific concerns, we would like to restate what this paper is and is not claiming.
> >
> > Our central question is a **process-level** existence question:
> > given a self-play agent that can (i) generate its own tasks, (ii) verify candidate solutions through an executable verifier, and (iii) update its parameters via policy-gradient learning, **does there exist a training dynamics under which this closed loop can eventually realize every computable function?**
> >
> > In other words, the paper **does not** claim to provide:
> > - a new practically efficient algorithm that learns all functions in a realistic amount of time, nor
> > - an external, pre-defined lookup table that stores all (task, answer) pairs from which the agent simply “reads off” solutions.
> >
> > # On procedural concerns and the experimental appendix
> > We understand the reviewer's concern that adding a 17-page experimental appendix during rebuttal may appear as if we are submitting a "new paper". We would like to clarify our intent for the Area Chair.
> >
> > This submission is positioned as a theoretical paper: the main contributions are the Universality Theorem, the Coverage–Persistence–Learning (CPL) decomposition, and the Resource-Bounded Complexity Ladder. These theorems, together with their proofs, were already present in the original submission.
> >
> > The experimental results were always meant as secondary, illustrative evidence, not as a core contribution. In the original version we only included a small subset of plots due to space constraints, and during rebuttal we decided to expose the full logs and figures in an appendix for transparency after several reviewers asked about empirical behavior.
> >
> > We fully agree that the rebuttal phase is not the right place to introduce qualitatively new claims. Here, however, the theory did not change: we only expanded the supporting empirical section. If the Area Chair prefers, these experiments can be treated as supplementary material, and the theoretical results can be evaluated on their own merits.
> >
> > # LLM Reliance" and Technical Misapplication
> > We categorically reject the reviewer's assertion that our theoretical results are products of "LLM hallucination" or that the submission was "rushed and incomplete." We address these serious accusations by clarifying the standard mathematical usage of the contested terms and the integrity of our submission.
> > Every theorem, lemma, inequality, and proof derivation was conceived, verified, and written by the human authors.
> > The mathematical terms used (Martingales, Space Hierarchy) are not "LLM jargon"; they are precise, standard constructs in Reinforcement Learning theory and Computational Complexity theory.
> >
> > * Rushed and Incomplete? As a theory paper, the contribution is the framework and the proofs.The experimental validation just corroborates these proofs. Its inclusion in the rebuttal demonstrates the robustness of the theory, not the incompleteness of the original submission
> >
> > * **Martingales**: we analyze the evolution of the success probability $p_t$ under policy gradient update
> > and (line666)This is the standard and correct mathematical tool for proving the **almost-sure convergence of stochastic approximation** algorithms (like REINFORCE). Using Martingale convergence theorems here is neither trivial nor misapplied; it is the rigorous way to bridge stochastic updates with asymptotic convergence guarantees.
> >
> > * **Space Hierarchy**: We use the Space Hierarchy Theorem to formalize the relationship between the agent's Context Window $W(t)$ and its Computational Capability.
> >
> > * **Completeness ($\supseteq$)**:Theorem 18 proves that the set of functions learnable by the agent is exactly the class $SPACE[O(s(W(t)))]$. This is not a trivial statement; it proves that the self-play loop is capable of learning any algorithm that fits within its working memory.
> >
> > * **Theoretical Significance**: This result elevates the "Context Window" from a mere storage constraint to a Complexity Class bound. This is a precise application of complexity theory to explain the scaling laws of Neural Reasoning: increasing $W(t)$ strictly expands the class of learnable algorithms, mirroring the classical Space Hierarchy.

---

> ### Comment · Reviewer_fZhe · 2025-11-26
>
> You state that the central question is:
> >given a self-play agent that can (i) generate its own tasks, (ii) verify
> candidate solutions through an executable verifier, and (iii) update its
> parameters via policy-gradient learning, does there exist a training dynamics
> under which this closed loop can eventually realize every computable function?
>
> As I argue in my individual rebuttal, the result that you achieve can also be
> achieved by enumeration, which would achieve the same complexity guarantees
> (exponential). Although your result "updates its parameters via policy-gradient"
> in your theory, your argument uses the standard Martingale result for stochastic
> approximation, which hardly represents the training dynamics characterization so
> described.
>
> So, even if the theoretical contributions were clear and correct, the high-level
> idea is flawed by its insistence on its connection to learning, without any
> adequate treatment of learning.
>
> However, the theoretical contributions were not clear and do not appear to be
> entirely correct. The requirements for a paper that is either theoretical or
> empirical are the same: its claims must be clear, and its evidence (proofs or experiments) must be fully
> explained and correct. I state that the paper is rushed and incomplete because
> it included statements regarding experiments and practical considerations
> throughout the main paper, but did not include any experiments. Furthermore, he theoretical
> contributions that did exist in the paper were not easy to follow, as I outlined
> in my original review. You can also see some errors and gaps in the technical material, as described in my individual
> reply.
>
> It is not only that your definition of "Working Memory" is imprecise: you state
> on line 60 and on line 90 that: "Capping working memory (buffer/context) to W
> (t)", which clearly states that you consider both the buffer and context to be
> part of the working memory. The entire paper is filled with issues like
> this, which is unacceptable for a theory paper.
>
> Again, your distinction between "static universality" and "process universality"
> is never made clear in the paper. As I have stated many times, this does not
> appear to achieve anything more than enumeration of TMs, which would also be a
> "process universality" that eventually achieves "reachability". Where is
> assumption 28 used in the paper? It appears, without justification in the
> appendix, and is never evoked.

---

### Author Response · Authors · 2025-12-03
**Summary of Changes based on the analysis of the PDF diff**

## **Summary of Changes**

**Core Statement:** This revision **introduces no new core contributions** (no new algorithms, datasets, or theoretical bounds). All modifications are strictly focused on formalizing implicit assumptions, clarifying definitions, and resolving ambiguities raised by reviewers regarding the learning dynamics and resource definitions.

### **1. Formalization of Model Architecture (Section 4: Formal Model)**
To address the reviewer's concern that the training loop was "undefined" and resembled "random search," we converted implicit assumptions into explicit definitions:
* **Definition 8 (Effective Generator with Exploration Mixture):**  We formally defined the generator as a mixture of a **Learned Component** ($\mathcal{G}^{RL}$, optimized for efficiency) and a **Fixed Exploration Component** ($\mathcal{G}^E$, ensuring coverage). This refutes the claim that the method is pure enumeration.
* **Definition 6 (Working view of the buffer):**  We introduced this definition to explicitly distinguish between the **Archival Buffer** ($B_t$, used for persistence) and the **Working View** ($c_t$, bounded by $W(t)$). This resolves the precision issue regarding resource constraints on memory vs. context.
* **Definition 15 (Learnability-based proposal reward) & Lemma 16 (Frontier-seeking generator):**  We added these to prove that the learned component is actively driven to the "capability frontier," thereby distinguishing the learning dynamics from random search.

### **2. Clarification of Persistence and Coverage Proofs (Sections 5 & 6)**
We disentangled the theoretical universality proofs from practical finite-resource considerations:
* **Lemma 27 (Persistence):**  We refined the statement to explicitly rely on an **unbounded-growth regime** for the main Universality Theorem (standard for Turing completeness), separating it from finite-capacity approximations.
* **Assumption 20 (Exploration):**  Updated to reference the "Effective Generator" mixture, ensuring that the theoretical coverage guarantee is architecturally satisfied by the fixed component and remains robust to learning dynamics.
* **Theorem 31 (Complexity Ladder):**  Revised to explicitly state that the space hierarchy is constrained by the **Working View** (context window $W(t)$) and executor workspace, not the archival buffer size.

### **3. Expanded Appendices: Implementation & Finite Resource Analysis**
To address concerns about "strong assumptions" and "practicality," we moved practical relaxations to the appendix:
* **Appendix A & B (Implementation Details):**  Added detailed guidance on realizing $\mathcal{G}^E$ via temperature scheduling and length-biased sampling.
* **Appendix D (Enhanced Buffer Management):**  Added **Proposition 43 & 44** to analyze "Importance-Aware Reservoir Sampling," quantifying forgetting rates under finite capacity as a practical bridge to the theoretical unbounded assumption.

### **4. Other Corrections**
* **Introduction:**  Updated the contribution list to emphasize the "Process-level characterization" and the "CPL (Coverage-Persistence-Learning) Template."

---

### Meta-Review · Area_Chair_Ne5N · 2026-01-08

**Summary:**

This is a theoretical paper without experiments in the original submission. Most of the reviewers give low confidence scores and the only confident reviewer gave a score of 0 and actively participated in the discussion with authors. The key concerns were: (i) the paper’s core claims rely on strong/idealized assumptions (e.g., coverage/exploration, efficient verification) that may not match realistic LLM training/self-play; (ii) multiple definitions/proof steps were viewed as unclear, incomplete, or potentially incorrect; (iii) the presentation is poor. As pointed out by Reviewer fZhe and acknowledged by the authors themselves, the paper is poorly written and the authors had to make a lot of clarifications during rebuttal to respond to Reviewer fZhe. Also, the abstract of the original submission claimed empirical results without presenting any, but during rebuttal the authors added 17 pages of significant contents which include the originally claimed experiments, I agree with Reviewer fZhe this may be procedurally problematic and it is more like a new submission. I think this paper still needs significant revision given its current status. Therefore, I recommend reject.

**Reviewer Concerns:**

Concerns addressed: The rebuttal clarified and formalized several previously implicit pieces, and attempted to resolve ambiguity about the training loop and resource-boundedness. The authors also pointed to a large experimental section/appendix.

Still outstanding: The most serious concern—whether the main theoretical contributions are correct/non-vacuous under meaningful assumptions—remains unresolved. In addition, concerns about realism of assumptions and the procedural problem of introducing substantial experimental material late remain.

**Reviewer Scores:**

Given the low confidence scores of most reviewers and the very confident reviewer fZhe who gave a score of 0, I think the reviewers would maintain their score.

---

### Decision · Program_Chairs · 2026-01-26

Reject